# RNA Polymerase II cluster dynamics predict mRNA output in living cells

Won-Ki Cho[1], Namrata Jayanth[1], Brian P English[2], Takuma Inoue[1],
J Owen Andrews[1], William Conway[1], Jonathan B Grimm[2], Jan-Hendrik Spille[1],
Luke D Lavis[2], Timothée Lionnet[2]*, Ibrahim I Cisse[1]*

[1]Department of Physics, Massachusetts Institute of Technology, Cambridge, United States; [2]Janelia Research Campus, Howard Hughes Medical Institute, Ashburn, United States

**Abstract** Protein clustering is a hallmark of genome regulation in mammalian cells. However, the dynamic molecular processes involved make it difficult to correlate clustering with functional consequences in vivo. We developed a live-cell super-resolution approach to uncover the correlation between mRNA synthesis and the dynamics of RNA Polymerase II (Pol II) clusters at a gene locus. For endogenous β-actin genes in mouse embryonic fibroblasts, we observe that short-lived (~8 s) Pol II clusters correlate with basal mRNA output. During serum stimulation, a stereotyped increase in Pol II cluster lifetime correlates with a proportionate increase in the number of mRNAs synthesized. Our findings suggest that transient clustering of Pol II may constitute a pre-transcriptional regulatory event that predictably modulates nascent mRNA output.

**\*For correspondence:** lionnett@ janelia.hhmi.org (TL); icisse@mit. edu (IIC)

**Competing interests:** The author declares that no competing interests exist.

## Introduction

The nuclear organization of eukaryotic cells can play important roles in the regulation of genome function (*Pederson, 2002*). In particular, the formation of sub-compartments is hypothesized to help increase the local concentration of essential enzymes and render nuclear processes more efficient (*Mao et al., 2011*). Enzymatic clusters have been reported in many genetic processes in the cell (*Carmo-Fonseca, 2002*; *Cremer et al., 2006*; *Schneider and Grosschedl, 2007*), including replication (*Kennedy et al., 2000*), DNA repair (*Misteli and Soutoglou, 2009*), transcription (*Cook, 2010*; *Verschure et al., 1999*) and RNA processing (*Kumaran et al., 2008*). These clusters can form *de novo* and their stability can be dynamically regulated in vivo making it difficult to capture them and to study their function with mechanistic detail (*Sutherland and Bickmore, 2009*; *Fraser and Bickmore, 2007*; *Buckley and Lis, 2014*).

In mammalian cells, the spatial organization of transcription has been revealed primarily with chemically fixed (non-living) cell techniques. These techniques include fluorescence in situ hybridization (*Femino et al., 1998*; *Mitchell and Fraser, 2008*; *Fraser and Bickmore, 2007*), immunostaining (*Iborra et al., 1996*), and chromosome conformation capture and immunoprecipitation-based approaches like 3C (*Tolhuis et al., 2002*; *Osborne et al., 2004*), HiC (*Lieberman-Aiden et al., 2009*), ChIA-PET (*Li et al., 2012*). Clusters of RNA Polymerase II (Pol II) were initially observed in fixed cells (*Jackson et al., 1993*; *Papantonis and Cook, 2013*) via anti-body staining against the active forms of the polymerase, and seen to co-localize with sites of nascent RNA synthesis in the fixed cells. From these fixed cells studies emerged theories interpreting the Pol II clusters as static pre-assemblies termed "transcription factories." However, attempts to directly visualize Pol II clusters in living cells had been initially unsuccessful (*Sugaya et al., 2000*; *Kimura et al., 2002*), raising a debate over their existence in vivo (*Carter et al., 2008*; *Sutherland and Bickmore, 2009*).

In earlier studies, limitations of conventional live-cell imaging methods may have contributed to the failure to detect non-homogeneous spatiotemporal organization of Pol II in living cells. Specifically, conventional imaging methods do not readily resolve substructures at length scales below the optical diffraction limit. Another difficulty arises if clusters exhibit fast kinetics. For instance clusters that form transiently may not be easily detectable. Capturing and understanding the spatiotemporal organization of Pol II in living cells can unveil hitherto hidden mechanisms for the regulation of gene expression in vivo.

Recent investigations of Pol II (*Cisse et al., 2013*) or an associated factor (*Ghamari et al., 2013*) in living cells, and new quantification in fixed cells (*Zhao et al., 2014*) revealed evidence for a highly dynamic Pol II cluster turnover process. The Pol II cluster dynamics (on the order of seconds) were significantly faster than the period required to complete the transcription of a typical mammalian gene (on the order of minutes) (*Cisse et al., 2013*). The lack of a correlative quantitative live-cell method, capable of capturing at high spatiotemporal resolution both the protein cluster and the transcriptional output, prevents further functional studies of Pol II clustering. For instance it is unclear whether transient protein clusters occur on actively transcribed genes, and whether the clustering event has a functional consequence on the gene expression process.

Here we develop a quantitative live cell, single molecule and super-resolution assay to capture protein clustering on an endogenous, actively transcribed gene. In live mammalian cells, the assay successfully co-localizes the polymerase clustering, in one color, with nascent RNA transcripts synthesized at the gene loci in a separate color. Our data reveal a previously uncharacterized, direct correlation between Pol II cluster lifetime and the number of nascent mRNA molecules subsequently synthesized. We find that this correlation between Pol II cluster lifetime and nascent mRNA output is predictive in nature, and may be utilized by an experimenter to stall or induce a burst of transcription, at will using a drug treatment. We discuss technical limitations as well as potential avenues for further studies on this largely uncharacterized mechanism for gene expression regulation.

## Results

### Quantitative super-resolution imaging

We set out to elucidate the spatiotemporal dynamics of Pol II in live mouse embryonic fibroblasts (MEF) using single-molecule based super-resolution microscopy (*Hess et al., 2006*; *Betzig et al., 2006*; *Rust et al., 2006*). In a photo-activation localization microscopy (PALM) approach (*Betzig et al., 2006*; *Hess et al., 2006*), for instance, individual fluorescently tagged proteins are photo-activated randomly while neighboring molecules remain dark or undetected. In our case, we photo-convert the fluorescent proteins (Dendra2) by illuminating the sample with very low intensity 405 nm light (1.3 W/cm$^2$).

Dendra2 (*Gurskaya et al., 2006*) is an initially green-emitting fluorescent protein that upon 405 nm illumination converts into a red-emitting form (*Chudakov et al., 2007*; *Zhang et al., 2007*). Dendra2's favorable photo-physics (*Lee et al., 2012*) and low aggregation propensity (*Landgraf et al., 2012*) make it optimal for our purposes. Fluorescence emitted in the converted state is detected using an EM-CCD camera. By fitting a Gaussian profile to the signal, one can measure the position of a molecule with a precision better than the diffraction limit of conventional fluorescence microscopes.

In our experiments (acquired with a 60 ms frame rate), only molecules that are immobile within this 60 ms temporal window are detected and represented in live cell super-resolution data; fast diffusive molecules are motion-blurred and are not represented. The observation of a molecule in a single frame is registered as a count of one "detection". The camera continuously records for typically 10,000 frames. The photo-converted Dendra2 molecules typically remain in the emitting state for tens to hundreds of milliseconds, and therefore a single molecule may appear in multiple frames and hence may be detected more than once. The fluorescent molecule then typically undergoes an irreversible transition into a permanent, dark state (photo-bleaching). However, the molecule may also transition to a transient dark state (blinking) before photo-bleaching. These intermittent photo-physical blinking transitions complicate direct conversion from counts of detections into exact numbers of molecules (*Lee et al., 2012*; *Rollins et al., 2015*). To avoid propagation of counting error, we use the raw number of detections as our metric in quantitative super-resolution analyses.

Additional challenges arise when quantitatively interpreting super-resolution data. The number of converted molecules at any given time can be controlled by the power density of 405 nm illumination. Ideally, sufficiently low 405 nm illumination is applied such that at most one molecule is detected per frame in a given region of interest. This sparse activation is a necessary condition for precise localization in single-molecule based super-resolution techniques. Consequently, at any given time, super-resolution techniques detect only a small subset of all fluorescent molecules present; and over a period of time, it is the frequency of detections that represents the relative local protein concentration (*Figure 1A*). Regions of high local protein concentration will have a higher frequency of detections compared to regions of low protein concentration. This is the principle behind the time-correlated detection counting method we developed to reveal the dynamics of protein clusters existing below the diffraction (*Cisse et al., 2013*).

We previously introduced time-correlated detection counting with PALM (tcPALM) as a powerful method to measure the spatiotemporal organization and dynamics of Pol II with single-molecule resolution (*Cisse et al., 2013*). Here we build on this approach, combining PALM and stochastic optical reconstruction microscopy (STORM) (*Rust et al., 2006*), orthogonally in vivo, to investigate the function of Pol II clustering during transcription at a specific gene locus, in living cells.

## Pol II clusters transiently in live mouse embryonic fibroblasts

We generated a mouse embryonic fibroblast (MEF) cell line stably expressing RPB1, the catalytic subunit of Pol II fused with the photo-convertible fluorescent protein, Dendra2. We used an α-amanitin resistant mutant of RPB1. To ensure that the functional Pol II proteins in the cells contained the Dendra2 construct, we treated the cells with α-amanitin to degrade endogenous Pol II. Although mainly localized in the nucleus, some Pol II may be detected in the cytoplasm due to the protein's translation, folding, or degradation processes. We choose to study only those Dendra2-Pol II signals that appear in the cell nucleus where transcription by Pol II takes place. In summary, we have created an MEF cell line where endogenous Pol II is replaced with Dendra2-Pol II to make it amenable to super-resolution imaging.

We obtained super-resolution images of Pol II in live cell nuclei (*Figure 1B*). High local concentrations of Pol II molecules appear as bright regions in the super-resolution images. We interpret the distinct bright foci as Pol II clusters (*Cisse et al., 2013*). We confirmed that the apparent spatial clusters reflected clustering of multiple molecules, rather than stochastic blinking of single molecules, by performing spatial pair-correlation PALM analysis as previously published (*Sengupta et al., 2011*) (see Appendix 1 and *Figure 1—figure supplement 1A–C*).

We select individual clusters from the live cell super-resolved data for tcPALM analysis (see scheme in *Figure 1A*). We first plot, as a function of time, the number of detections per frame (*Figure 1C,D* and *Figure 1—figure supplement 1D*). Additionally, we represent the same data as a cumulant, which is the sum of all detections counted from beginning of acquisition (time t = 0) up to that point in time. We refer to these temporal representations as a cluster's "time trace". The time traces of representative Pol II clusters show evidence for rapid cluster assembly characterized by a steep, sudden inflection in the slope of the cumulant, arising from an increase in the frequency of detections (black arrows in *Figure 1C,D*). We interpret as *de novo* cluster assembly, events in which the onset of slope increase occurs after the acquisition has started (i.e. after time t = 0). Steep slopes are followed by an abrupt plateau, due to a sudden decrease in detection frequency with virtually no Pol II detection events. We interpret the inflection points with sudden transitions from very high to very low frequency of detections as cluster disassembly events. The interpretations of the time trace are corroborated by Monte Carlo simulations of the tcPALM method (*Figure 1—figure supplement 2*).

The temporal analyses of Pol II data in live MEFs (*Figure 1C,D* and *Figure 1—figure supplement 1D*) show transient Pol II clusters. In contrast to *de novo* assembly, the signatures of stable Pol II clusters in fixed cells (*Figure 1—figure supplement 1E*) feature a large positive slope from the start of acquisition, indicating existence of the cluster before acquisition. The initial linear increase is followed by a more gradual transition to a plateau, reflecting the gradual depletion of the pool of unconverted Dendra2-Pol II through continued photo-conversion. To further verify that the distinct temporal signatures between fixed and live cell data did not stem from the fixation procedure, we decided to image a stable protein cluster in live cells. We generated a fusion of the histone H2B, well known to remain stably bound within the short timescales of our experiments (*Chen et al.,*

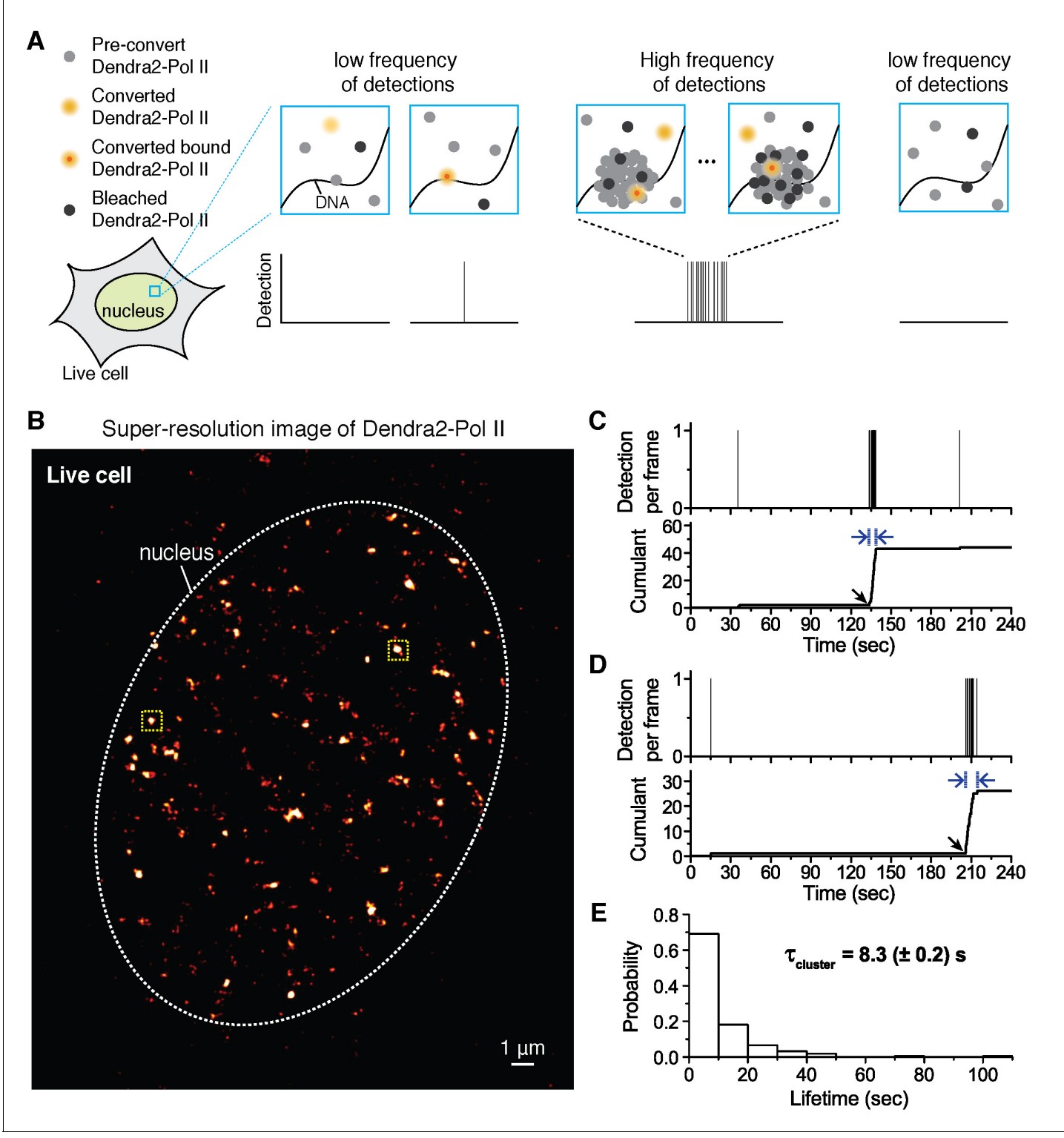

**Figure 1.** Quantitative super-resolution imaging unveils transient pol ii clustering in live mouse embryonic fibroblasts (MEF). (**A**) Schematic representation of time-correlated detection counting: individual molecules that are photo-converted and bound within 60 ms frame rate are detected; changes in local protein concentrations are evidenced by relative changes in the frequency of detections (**B**) Super-resolution reconstruction of Dendra2-Pol II in live MEF, depicting clusters of Pol II. Red-hot color code denotes spatial density of detections accumulated from 5000 frames. (**C, D**) Representative time traces for two selected Pol II clusters. The selected regions are shown as yellow squares in **B**. In the traces, time t = 0 represents the start of acquisition. Black arrows indicate the onset of Pol II clustering, and blue arrows indicate the apparent cluster lifetime. (**E**) A histogram of the
*Figure 1 continued on next page*

*Figure 1 continued*

apparent cluster lifetimes ($\tau_{cluster}$) is presented for 214 clusters pooled from 6 live MEF cells. Globally, throughout the nucleoplasm of live MEF cells an average lifetime, $\tau_{cluster}$, of 8.3 (± 0.2) s was obtained. Errors (in parentheses) represent standard error of the mean.

The following source data and figure supplements are available for figure 1:

**Source data 1.** Source data for 1D.

**Figure supplement 1.** Spatial clustering of Pol II determined by pair-correlation (pcPALM) analysis, A zoom-in trace of transient Dendra2-Pol II cluster in live cells, A trace of Dendra2-Pol II in fixed cells, and traces of H2B-Dendra2 in live cells.

**Figure supplement 2.** Monte-Carlo simulation of Pol II cluster dynamics as detected in tcPALM.

**Figure supplement 3.** Pol II cluster lifetime is independent of 561 nm excitation laser power.

*2014*; *Izeddin et al., 2014*) labeled with our photo-convertible protein tag (H2B-Dendra2). Imaged in identical conditions as Pol II, H2B-Dendra2 proteins displayed the gradual monotonic slopes from the beginning of acquisition expected for tcPALM analysis of stable structures, even when imaged in living cells (see Appendix 2 and *Figure 1—figure supplement 1F,G*). This confirms that time-correlated super-resolution analysis is readily able to discriminate between stable and transient protein assemblies. To summarize, the live cell time traces of Pol II clusters in MEF are not consistent with stable clusters, and rather reflect clusters that are rapidly assembled and disassembled.

From the observed time traces, we define the apparent cluster lifetime ($\tau_{cluster}$) as the duration from slope onset to the subsequent plateau in the cumulant graphs. We analyzed time traces of 214 Pol II clusters pooled from 6 live MEF cells under normal growth conditions, and obtained an average cluster lifetime of 8.3 (± 0.2) s (*Figure 1E* and *Figure 1—source data 1*). The average cluster lifetime is independent of the excitation laser power, corroborating the conclusion that the measured lifetime is due to transient cluster dynamics and not single molecule photo-bleaching (*Figure 1—figure supplement 3*). Thus, under normal growth condition, the Pol II clusters in MEFs are very short-lived. The observed seconds-long lifetime of Pol II clusters in the murine cell line is comparable to our earlier study showing a Pol II cluster lifetime of 5.1 (± 0.4) s in human cells (*Cisse et al., 2013*).

## Capturing Pol II clustering at active β-Actin gene loci in living cells

We tested whether transient Pol II clusters are attributable to active transcription at specific gene loci, or are merely randomly occurring events. We hypothesize that if transient Pol II clusters play a role in transcription regulation, then they will co-localize with an actively transcribed gene in the nucleus of living cells. We therefore decided to label the messenger RNA of β-actin, a gene with well-characterized transcriptional dynamics (*Lionnet et al., 2011a*).

To enable the co-visualization of Pol II clusters and fluorescent β-actin mRNAs, the Dendra2-Pol II MEF cell line (used in *Figure 1*) was engineered from a mouse where a tandem repeat of a sequence derived from the bacteriophage MS2 had been knocked into the endogenous β-actin gene (*Lionnet et al., 2011a*). Upon transcription, the inserted MS2 sequence forms a hairpin structure in the untranslated region of the mRNA. This hairpin is specifically recognized by the coat protein of bacteriophage MS2 (MS2 Coat Protein, MCP), which is exogenous to mammals (*Bertrand et al., 1998*). When the MEF cells stably express MCP fused to the self-labeling HaloTag, the MCP-HaloTag molecules specifically decorate and accumulate on the tandem hairpin structure contained in the β-actin mRNA. The accumulation of MCP-HaloTag on the mRNA enables imaging of individual β-actin mRNA molecules (*Lionnet et al., 2011a*). To visualize the MCP-HaloTag proteins, we use a far-red organic dye, Janelia Fluor 646 (JF$_{646}$) (*Grimm et al., 2015*) conjugated to a HaloTag ligand. The JF646-HaloTag ligand readily penetrates living cells to covalently bind to the MCP-HaloTag protein and is spectrally distinguishable from the Dendra2 fluorescent protein labeling Pol II.

When multiple nascent mRNAs are synthesized at the β-actin gene loci, a bright accumulation of MCP-HaloTag signal is observed at the gene loci. Using the fluorescence brightness and mobility analysis to discriminate between single mRNA signal and gene loci with multiple mRNAs

(*Lionnet et al., 2011a*), we can employ the MS2/MCP fluorescent reporter system to find the position of the endogenous β-actin gene loci. We can also use the fluorescence to count the number nascent β-actin mRNA molecules observed at each gene locus.

In response to serum induction (see Materials and methods), prominent foci of $JF_{646}$ fluorescence are observed (*Figure 2A* and *Figure 2—figure supplement 1A*), consistent with sites of nascent RNA transcription as previously described (*Lionnet et al., 2011a*). In living cell nuclei, we observe up to four such loci (*Figure 2—figure supplement 1B*); this is consistent with the previous observation that the immortalized MEF cells are tetraploid. The Dendra2-Pol II expression therefore did not change the MS2 knocked-in β-actin gene levels in any apparent way (*Figure 2—figure supplement 2*), nor did it affect transcription dynamics in response to serum (*Lionnet et al., 2011a*; *Kalo et al., 2015*).

In our experiments, we focus on a single plane containing at least one β-actin gene locus (*Figure 2—figure supplement 1C*), and acquire live cell super-resolution STORM images (*Rust et al., 2006*; *Heilemann et al., 2008*) to better localize the β-actin gene locus (*Figure 2C*). A major advantage of live-cell STORM is that all the fluorescent molecules are detected within a short window of acquisition, such that the total fluorescence readily correlates with the number of mRNA molecules. For Pol II clusters, on the other hand, the low activation in a PALM approach filters out a large background Pol II to reveal the transiently clustered proteins. We combine STORM on the mRNA signal and PALM on the Pol II signal to reveal, respectively, the gene locus and any Pol II recruitment at the same locus, with high resolutions in living cells.

We then subject the same plane in which we initially detected a site of beta-actin transcription to tcPALM in order to image Dendra2-Pol II. We observe Pol II clusters throughout the cell nucleus (*Figure 2B and D*) consistent with our observations in *Figure 1*. Pol II clusters exist at several distinct loci, and using our β-actin mRNA STORM signal as a beacon for the actin locus, we can assay whether clusters occur specifically at the β-actin gene.

Merging the two super-resolved images, we observe that Pol II clusters co-localized with the β-actin gene (*Figure 2E*). The localization accuracies measured based on the detected Dendra2 and $JF_{646}$ fluorescence are 31 nm and 18 nm respectively (see Appendix 3 and *Figure 2—figure supplement 1D–I*). Within our super-resolution localization accuracy, we conclude that Pol II molecules do cluster on the actively transcribed β-actin gene in living cells.

We then analyzed specifically the Pol II clusters co-localizing with the transcribed β-actin gene. As illustrated by the example in *Figure 2F*, the Pol II clusters are short-lived (6.3 s in the example represented), consistent with our global observation of short-lived Pol II clusters throughout the nucleus (as in *Figure 1E*). Therefore transient Pol II clusters occur, at least in part, on active gene loci.

## Pol II clusters are more numerous than polymerases actively transcribing the gene

The identification of Pol II clusters on the β-actin gene locus provides an excellent basis to test the functional relevance of Pol II clusters. Our observation of short-lived clusters is not consistent with elongating polymerases: if Pol II clusters represented elongating polymerases on the gene, we would expect them to last several minutes on the β-actin gene since each Pol II molecule takes several minutes to transcribe a single mRNA. We investigate further the putative difference between Pol II clusters and elongating Pol II by counting the number of molecules detected in a cluster.

To estimate the number of Pol II molecules in a cluster, we considered a limitation in the live cell super-resolution that can be addressed in fixed cells: due to the low photo-conversion rate, our live-cell super-resolution assay may be blind to the smaller number of polymerases actively elongating throughout the gene (scheme in *Figure 1A*). In particular, clusters may disassemble faster than it would take the live cell super-resolution method to photo-convert and detect all the molecules present in a cluster. We therefore reasoned that chemically fixing the cells may reveal two populations of co-localizing Pol II molecules: a population with smaller number of polymerases attributable to elongating Pol II, and a second population of Pol II clusters that are numerically bigger and distinct from elongating Pol II.

We use paraformaldehyde to chemically crosslink proteins in a way that minimally disrupts the spatial distribution of biomolecules and in the hope to freeze transient clusters. We then performed the same dual-color super-resolution imaging as in *Figure 2* but this time in fixed cells. Represented in *Figure 3* is data for cells fixed 12 min after serum stimulation. In *Figure 3A*, two loci are detected

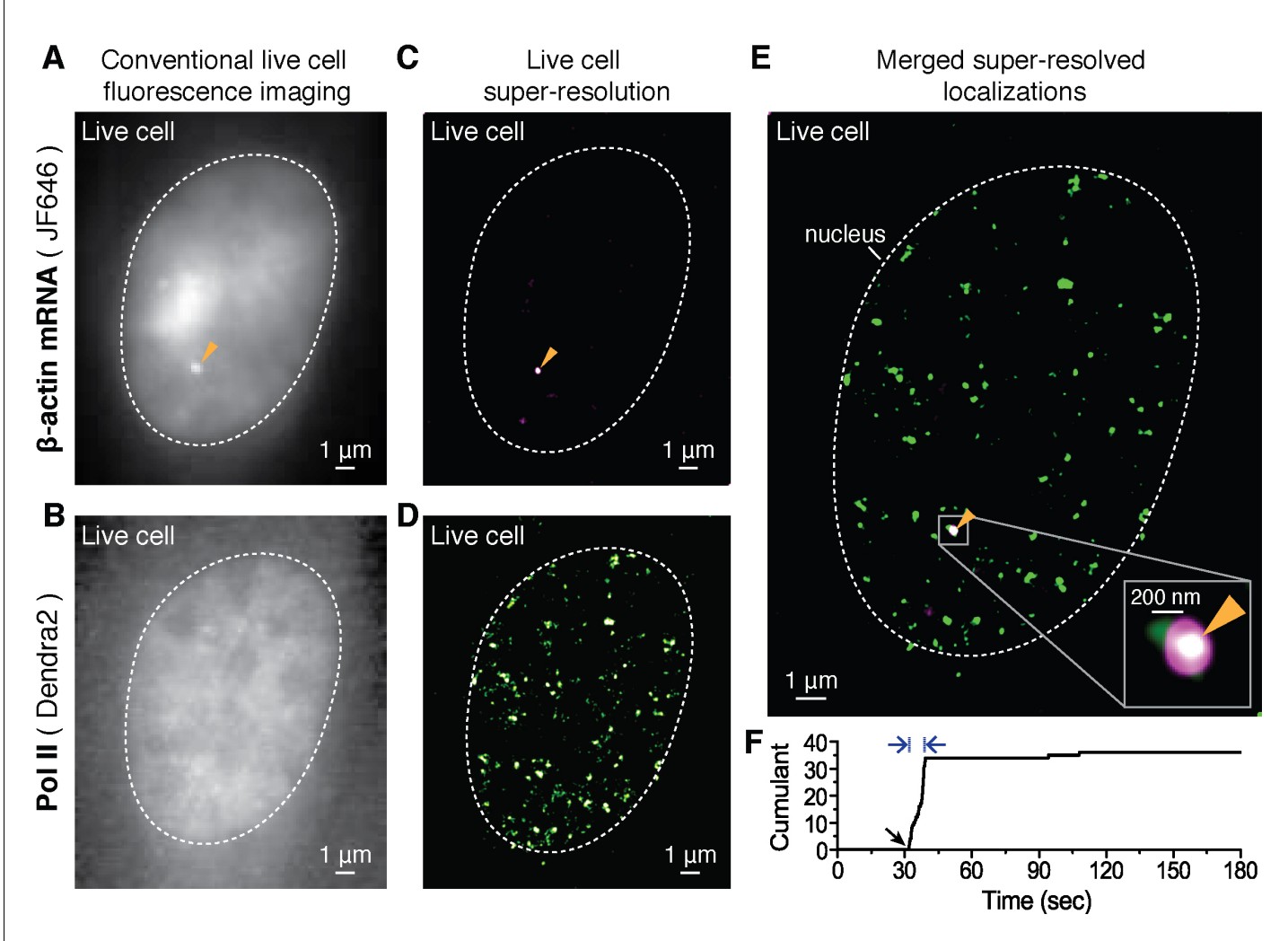

**Figure 2.** Dual-color super-resolution imaging captures Pol II clustering at active β-actin gene locus in a living cell. In a live MEF cell, endogenous β-actin loci can be observed (orange arrows in **A**, **C** and **E**). (**A**) Conventional fluorescence image of JF$_{646}$-labeled β-actin mRNA and (**B**) pre-converted (conventional) Dendra2- Pol II imaging, at the same imaging plane as in **A**, in the same live cell nucleus. The images are acquired for a plane focusing on a bright JF$_{646}$-β-actin gene locus (orange arrow in **A**). The images are averaged projections from 100 frames in **A** and 300 frames in **B**, respectively. (**C**) Super-resolution (STORM) reconstruction is performed to localize the β-actin gene locus with high precision; a magenta-hot color code is used to represent the spatial density of labeled mRNA detections accumulated from 200 frames. Because most of the mRNA is diffusing throughout the nucleoplasm of the living cell, mainly the immobile nascent mRNA signal appear as a clear signal in the super-resolved image (**D**) Super-resolution (PALM) reconstruction of converted Dendra2-Pol II is performed to image the Pol II clusters in the same plane of focus. A green-hot color code is used to represent density of labeled Pol II detections. (**E**) Merging the two super-resolved images of the β-actin gene locus (magenta) and Pol II (green) reveals the co-localization of a Pol II cluster with the active gene locus. Standard magenta and green are used for β-actin mRNA and Pol II respectively, such that co-localization should appear as white. (**F**) The live cell super-resolution time trace for the Pol II cluster co-localizing with the active gene locus shows an apparent cluster lifetime of 6.3 s, consistent with the transient clusters observed generally throughout live MEF cell nuclei (*Figure 1E*). Black arrow indicates the onset of Pol II clustering, and blue arrows indicate apparent lifetime. The cell was imaged 5 min after serum induction (additional co-localization data at different times after serum induction are shown in *Figure 4* and *Figure 4—figure supplement 2*).

The following figure supplements are available for figure 2:

**Figure supplement 1.** Identification of nascent transcription loci and localization precision of Dendra2 and JF$_{646}$ as calculated from the live cell super-resolution data.

**Figure supplement 2.** Quantification of β-actin mRNA copy number per cell (using smFISH) in the cell line expressing Dendra2–Pol II, compared to the cell line not expressing Dendra2–Pol II.

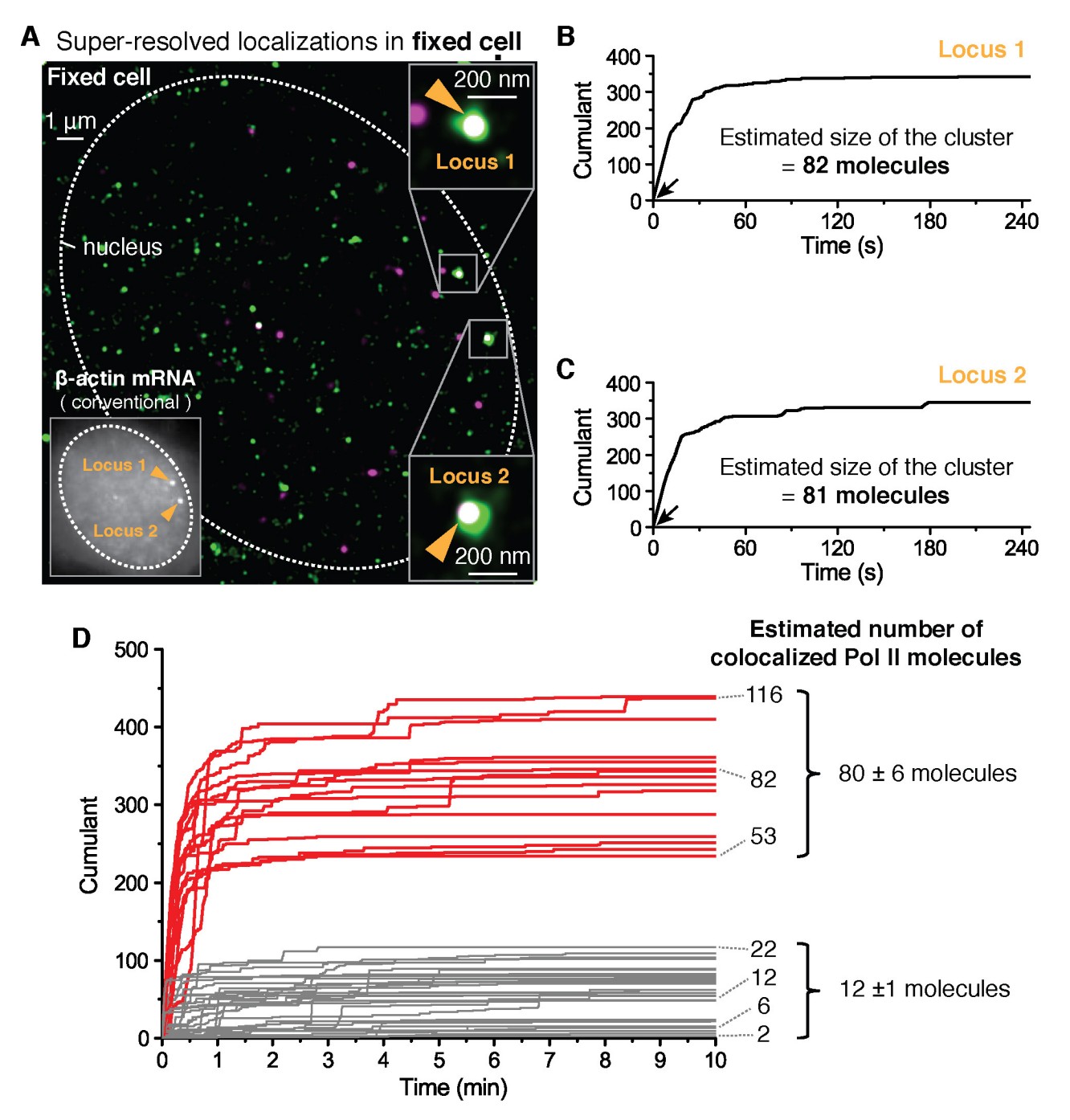

**Figure 3.** Fixed cell super resolution analyses reveal Pol II clusters are more numerous than polymerases actively transcribing the β-actin gene. (A) The merged, super-resolved image of Dendra2-Pol II (green, PALM) and $JF_{646}$-β-actin mRNA (magenta, STORM) is presented for a MEF cell chemically fixed 12 min after serum stimulation. (A, left inset) conventional fluorescence image shows two bright gene loci (orange arrows, indicated as "Locus 1" and "Locus 2") in the same plane of focus for the same cell. In a fixed cell background non-diffusing mRNA may also appear as puncta of lower intensity in both conventional, and STORM images. (A, right insets) Co-localized Pol II clusters are detected on both gene loci. Standard magenta and green are used for β-actin mRNA and Pol II respectively, such that co-localizations appear as white. (B, C) The cumulant graphs for the two co-localized Pol II clusters are represented for Locus 1 and Locus 2, respectively. Note the slope onset (black arrows) from the start of acquisition indicative of stable clusters as expected for fixed cells. Both cumulant graphs feature total counts greater than 200 detections. The estimation of the count of Dendra2-Pol II molecules in the cluster is ~80 molecules for each locus. (D) Cumulant graphs of co-localized Pol II signals from 34 fixed cells are represented. More frequent are Pol II accumulations with relatively low counts of detections (in gray, 29 out of 44 cumulants), and molecule counts consistent with the number of Pol II engaged in β-actin mRNA synthesis (see *Figure 3—figure supplement 1H*). These may likely represent elongating Pol II. Also featured

*Figure 3 continued on next page*

*Figure 3 continued*

in the cumulant graphs are large Pol II clusters (in red, 15 out of 44 cumulants) with total counts greater than 200 detections, and molecular counts significantly higher than expected for elongating Pol II on a β-actin gene. These likely represent the rare transient clusters observed in living cells, captured during the fixation process. For the estimated number of Dendra2-Pol II molecules for a selection of Pol II cumulants, see Appendix 4,5 and *Figure 3—figure supplement 1*. Temporal localization profiles for 44 Pol II clusters co-localized with β-actin gene loci are available in *Figure 3—source data 1*.

The following source data and figure supplement are available for figure 3:

**Source data 1.** Source data for 3D.
**Figure supplement 1.** Estimation of cluster size with photo-physical parameters of Dendra2.

in the same imaging plane, both co-localizing with apparently large clusters of Pol II (as estimated by the number of detections in the cumulant) (*Figure 3B and C*). Plotting the cumulant graphs of Pol II co-localized with gene loci from 34 fixed cells shows a clear separation between clusters with large counts of detections (red in *Figure 3D*, presumed as clusters large enough for short-lived detection in live cells) and those with smaller counts of detections (gray in *Figure 3D*).

To quantify the number of molecules in the clusters detected in fixed cell data, we first applied a previously reported computational approach, based on a hidden-Markov model that takes into account the Dendra2 photo-physics (*Rollins et al., 2015*) (see Appendix 4 and *Figure 3—figure supplement 1*). The algorithm converges to an extremum in the likelihood that a given cumulant graph originated from a number N of individual Dendra2-labeled biomolecules. This approach successfully converged for the cumulant graphs (gray in *Figure 3D*) with small counts of detections and we obtained estimates ranging from 1 to 20 molecules. However, the hidden-Markov approach failed to converge in finite time due to the large number of molecules for the bigger Pol II clusters (red in *Figure 3D*). The implementation of the hidden-Markov model had thus far counted complexes on the order of ~10 molecules, and by inspection of the cumulant graphs, we expect the large clusters to be from 5 to 10 times larger.

Alternatively, we developed a simple test that uses the same Dendra2 single-molecule photo-physical parameters (see Appendix 5 and *Figure 3—figure supplement 1A–D*), but also simulates multiple clusters of varying sizes and tests the best descriptors of the observed data (*Figure 3—figure supplement 1E*). This approach gave a range of 2–22 for the elongating polymerases, in good agreement with the hidden-Markov approach and consistent with the number of elongating polymerases previously reported for serum stimulated β-actin gene loci (*Lionnet et al., 2011a*; *Femino et al., 1998*; *Kalo et al., 2015*). The approach successfully approximated the large clusters to be on average ~80 Dendra2-Pol II molecules. In addition we observed a correlation between the smaller Pol II accumulations and the corresponding number nascent RNAs instantaneously detected at the gene locus (*Figure 3—figure supplement 1H*), while the correlation breaks down for the larger clusters.

We conclude based on the fixed cell co-localization data that the short-lived clusters observed on the β-actin gene loci are transient agglomerations of ~80 polymerases, which are distinct from the much lower counts of elongating polymerases. We expect that the lower counts of elongating polymerases likely appear as low frequency of detections as illustrated in *Figure 1A*.

## Pol II clustering lifetime correlates with number of nascent β-Actin mRNA

We investigated whether there is any relation between Pol II clustering and mRNA synthesis. The β-actin gene is known to display a stereotypical transcription pulse, with a peak output around 15 min after serum stimulation (*Lionnet et al., 2011a*; *Kalo et al., 2015*). We therefore examined whether the co-localized Pol II clusters exhibited similar changes associated with the gene activity.

Changes in Pol II cluster dynamics would exhibit signatures in the time traces which are quantitatively measurable with tcPALM analysis (*Figure 4—figure supplement 1A*). By analyzing the cluster time traces, we first identify whether the apparent cluster lifetime changes as a function of time after serum stimulation (*Figure 4—figure supplement 1B*). We also determine whether cluster incidences

occur more frequently (*Figure 4—figure supplement 1C*), or if the apparent local concentration of Pol II (as measured by the slope in the cumulant graph) changes as a function of time after serum stimulation (*Figure 4—figure supplement 1D*). We found that the apparent lifetime of Pol II clusters on the β-actin gene loci increases when observed in the interval between 10 and 15 min, compared to other periods after serum stimulation (*Figure 4* and *Figure 4—figure supplement 2*).

We attempted to follow a single gene locus for a long period to measure how clustering events on that locus differ as a function of time after stimulation. Unfortunately, however, imaging the same cell caused significant fluorescence photo-bleaching of both Pol II and the mRNA signals within a few minutes. The transcriptional activation of the β-actin gene in MEF cells after serum starvation has a well-characterized stereotyped response that remains evident after averaging multiple cells (*Lionnet et al., 2011a*; *Femino et al., 1998*; *Oleynikov and Singer, 2003*; *Kalo et al., 2015*). Here we average data from individual cells selected at varying times after serum stimulation, and each cell is imaged for a short period of up to 5000 frames (300 s). Cells were selected such that we obtained clustering dynamics data covering up to one hour after serum stimulations (*Figure 5A* and *Figure 5—source data 1*).

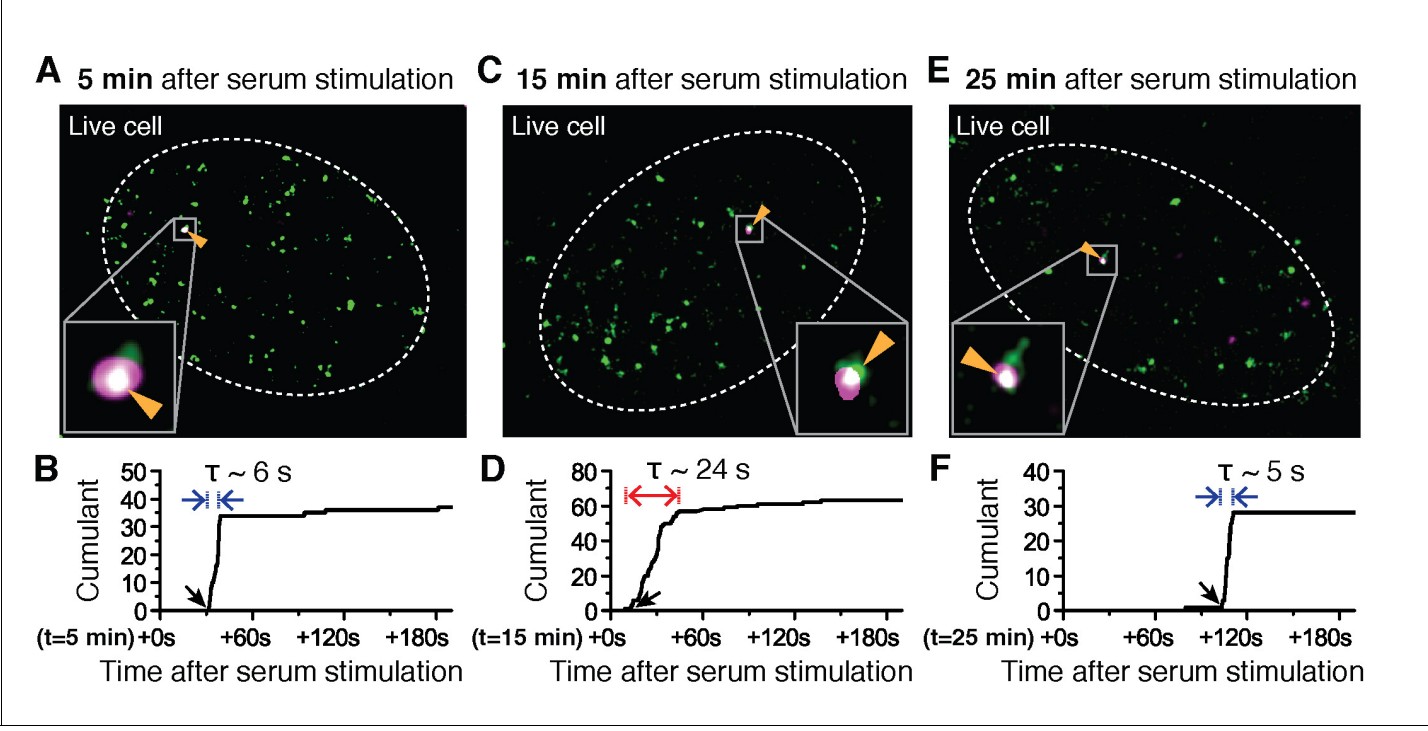

**Figure 4.** Super-resolved co-localization images of Dendra2-Pol II and JF$_{646}$-β-Actin mRNA at 5, 15 and 25 minutes after serum-induction. (A–C) Illustrated are three representative cells imaged at different time points after serum stimulation. Each cell shows co-localization between the pre-selectedβ-actin gene locus and a Pol II cluster in the merged super-resolved image. Magenta and green are used for β-actin mRNA and Pol II respectively, such that co-localization appears as white (D–F) Cumulant graphs of the Pol II clusters co-localizing with the nascent gene loci are represented. Consistently, cells imaged around 10 to 15 min after serum stimulation show increased cluster lifetime (red arrow in E), compared to clusters occurring outside that time window (blue arrows in D, F). The apparent lifetimes of the co-localized Pol II clusters are 6.3 s (D, blue arrows), 24.2 s (E, red arrows) and 4.8 s (F, blue arrows), for the cells imaged respectively at 5, 15 and 25 min after serum stimulation (additional examples at different times after serum induction are shown in *Figure 4—figure supplement 2*).

The following figure supplements are available for figure 4:

**Figure supplement 1.** Various cluster modulation models.

**Figure supplement 2.** Examples of dual color super-resolved co-localization images of Dendra2-Pol II and JF$_{646}$-β-actin mRNA at different time after serum-induction.

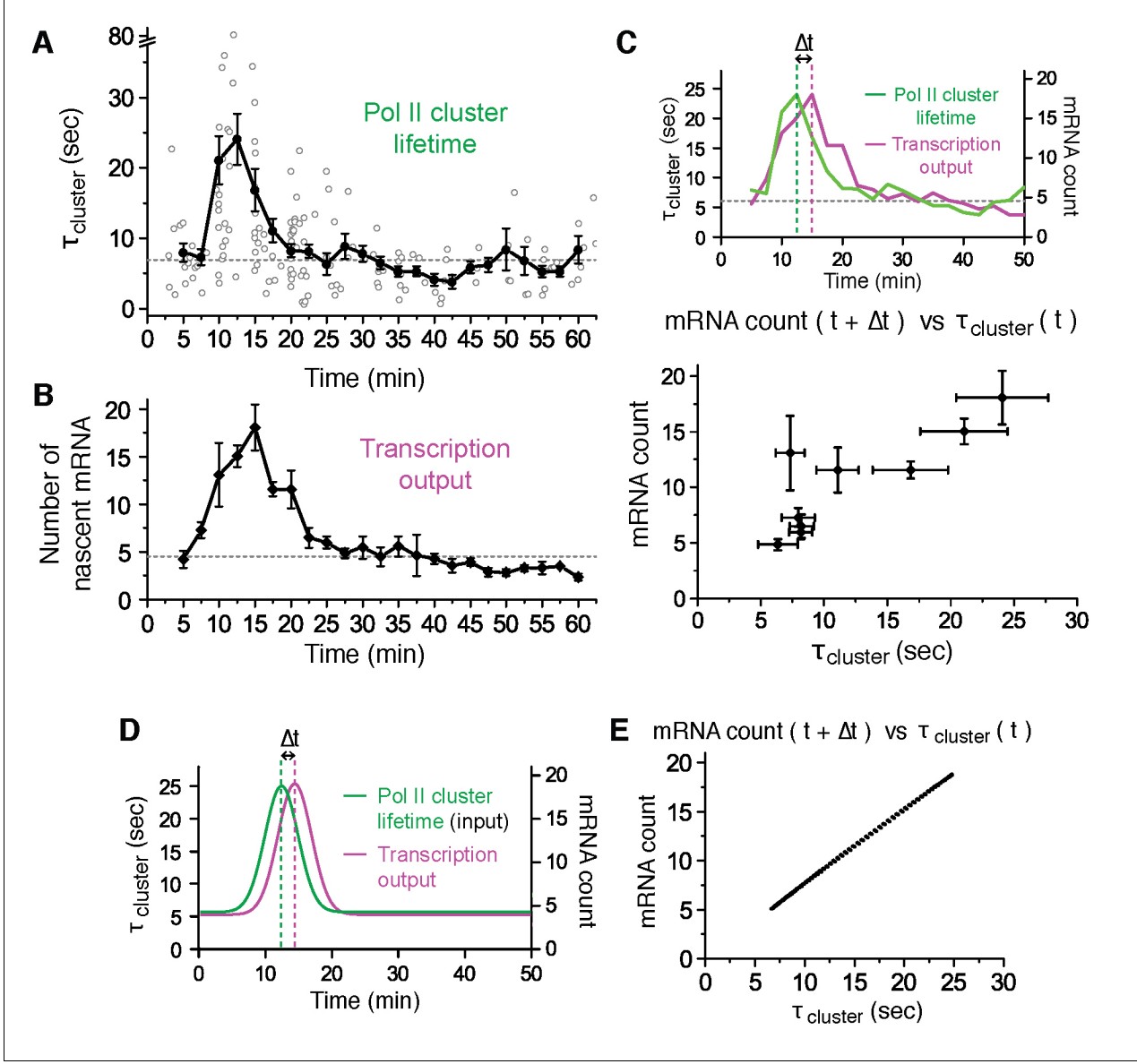

**Figure 5.** Pol II clustering lifetime correlates with the number of nascent β-Actin mRNAs in living cells. (**A**) Average lifetime for Pol II clusters co-localizing with nascent β-actin gene loci are plotted as a function of time after serum induction. The gray circles represent individual clustering events. 160 clusters from 89 dual-color super-resolved living cells were used in this graph. Horizontal dashed line represents the median of Pol II cluster lifetimes (6.9 s) from all the 160 clusters. The black dots are binned average lifetime from the pool of individual clusters. A peak Pol II clustering lifetime of 24 (± 3) s appears 12.5 min after serum stimulation. For each binned data point (black dots), we averaged N = 4 to N = 31 clusters (gray circles) co-localizing on the β-actin gene locus in a living cell imaged at the indicated time after stimulation. Error bars represent standard error of the mean. (**B**) Time-dependent plot of the average number of nascent mRNA per β-actin gene locus. A peak in mRNA output of 18 (± 3) nascent transcripts appeared 15 min after serum stimulation. Horizontal dashed line represents median of nascent mRNA outputs (4.5 transcripts). For each data point we averaged nascent mRNA counts from a minimum N=4 living cells imaged at the indicated time after stimulation. Error bars represent standard error of the mean. (**C**) Top: Time traces from panels **A** and **B** are overlaid to illustrate an observed time lag (delay Δt) between the peak Pol II cluster lifetime (green) and the peak mRNA output (magenta). Bottom: The average number of nascent mRNA (offset by the experimentally observed delay Δt = 2.5 min) is plotted against the Pol II cluster lifetime. Data for the first 30 min representing the response period are used; remainder of the data representing the basal level (from 30–60 min in **A** and **B**) are plotted in *Figure 5—figure supplement 1C*. Error bars represent standard errors on the mean. (**D, E**) An input-output model may account for the linear correlation between Pol II cluster lifetime and mRNA output. (**D**) Pol II cluster lifetime used as model input (green) is overlaid with a plot of the best fit for the mRNA output (magenta) from theoretical model. A time lag (delay Δt) is observed in model. (**E**) Relationship between the model output for the average number of nascent mRNA (offset by the model delay Δt = 2 min), plotted against the model cluster lifetime input recapitulates a linear correlation analogous to our experimentally observed data in C (experimentally observed delay was Δt = 2.5 min, and best-fit model delay was Δt = 2 min).

*Figure 5 continued on next page*

*Figure 5 continued*

The following source data and figure supplements are available for figure 5:

**Source data 1.** Source data for 5A, B and D.

**Figure supplement 1.** Time-dependent plots of spatial density and frequency of Pol II cluster, and a correlation Plot between number of mRNA and Pol II cluster lifetime 30–60 min after serum-stimulation.

**Figure supplement 2.** Stochastic model of Pol II clustering dependent transcriptional output.

We imaged 207 living cells, and found 89 living cells with Pol II clusters co-localized with the β-actin gene locus within the 300 s imaging window. The number of co-localization events we observed is consistent with a frequency of one Pol II clustering event every 5 min per β-actin locus (see *Figure 5—figure supplement 1B*). Thus the short imaging window, imposed by photo-bleaching, limits the likelihood of cells showing co-localized Pol II clusters on a gene locus. Nevertheless, the data from these 89 cells were sufficient in both temporal coverage as well as statistical significance to reveal a clear modulation of Pol II cluster dynamics in response to serum stimulation.

We find that upon serum stimulation, Pol II cluster lifetime increases 3 to 4-fold from a basal lifetime of ~6.9 s, to an average peak lifetime of 24 ± 3 s (at 12.5 min after stimulation), before relaxing back to the basal lifetime (*Figure 5A*). Similarly, we estimated the number of nascent mRNAs being synthesized at the transcription sites as a function of time after serum stimulation (*Figure 5B* and *Figure 5—source data 1*). We used the fluorescence intensity of single mRNA molecules diffusing in the nucleus as a reference to estimate the number of labeled mRNA molecules in the bright nascent transcription loci (as previously described in *Lionnet et al., 2011a*).

As represented in *Figure 5B,* we find that the mRNA synthesis burst peaks (around 15 min after serum stimulation) with an average of 18 ± 3 nascent transcripts per gene locus, before decreasing back to the basal loading level of ~4 to 5 transcripts. The burst response to serum stimulation, including the peak time and the counts of nascent mRNAs, is in agreement with previous studies of β-actin gene output (*Femino et al., 1998*; *Lionnet et al., 2011a*; *Kalo et al., 2015*).

We estimated the number of Pol II molecules per cluster as a function of time after serum addition, and it remained unchanged throughout serum response (*Figure 5—figure supplement 1A*). As a proxy for the number of Pol II molecules accumulated per transient clustering event, we define the clustering strength as the number of Pol II detections per unit time within the duration of a cluster (*Figure 4—figure supplement 1A*). This metric provides information on the relative spatial density of Pol II molecules in a cluster without the need for exhaustive counting of all molecules present, which is inherently challenging due to the short cluster lifetime. In addition to observing a flat cluster strength, we also observed a constant frequency of clustering events corresponding to ~0.2 detected clusters per minute, which remains unchanged up to an hour after serum stimulation (*Figure 5—figure supplement 1B*). Taken together, these data indicate that neither the number of Pol II molecules per clusters, nor the frequency of clustering is modulated in response to serum; only the lifetime of Pol II clusters was modulated in a manner correlated with gene expression output.

Thus, in response to serum stimulation the Pol II cluster lifetime and mRNA output were modulated from a basal value to a peak value 4-fold higher, followed by a relaxation back to the basal value. We note that there is an experimentally observable delay between Pol II cluster peak (around 12.5 min after stimulation) and mRNA output peak (around 15 min after stimulation). This delay (Δt) is consistent with the expected delay between Pol II recruitment and transcription the MS2 sequence ~3.4 kilobase pairs from the transcription start site of the β-actin gene.

We plotted the average Pol II cluster lifetime (at time t) against the number of nascent mRNA subsequently transcribed (at time t+Δt). For the data comprising the β-actin gene response to serum (i.e. the first 30 min after serum stimulation), we observe a direct, linear correlation between Pol II cluster lifetime and the number of nascent mRNA subsequently synthesized (*Figure 5C* and *Figure 5—figure supplement 1C*). In conclusion, we observe that Pol II cluster lifetime is directly correlated with the number of nascent mRNAs observed after an apparent lag time Δt.

## A stochastic model based on an input-output mechanism recapitulates our live cell observations

Our experimental results suggest a linear input-output relation between Pol II cluster lifetime and transcriptional output at the gene locus. To test the extent to which such a simple input-output relationship might account for our live cell measurements, we constructed a theoretical model of our system (see Appendix 6 and *Figure 5—figure supplement 2*). The model uses Pol II cluster lifetime as input. It makes the assumption that during a clustering event multiple Pol II can be loaded onto the gene, and the loaded Pol II molecules transcribe nascent mRNAs at a constant elongation rate. The elongation rate of Pol II is left as a free parameter such that the theoretical model can be fitted to the experimental data. We validate the model by comparing the rate parameters obtained from the fit to previously reported transcriptional rates.

The parameters obtained by fitting (see Appendix 6), suggest that our model recapitulates a delay (*Figure 5D*) and linear correlation (*Figure 5E*) between Pol II cluster lifetime and β-actin gene response (*Figure 5—figure supplement 2*). The lag time obtained in the model is Δt = 2 min, compared to the experimentally observed Δt = 2.5 min. The model yields a best fit Pol II elongation rate of ~3.1 kbp/min, in good agreement with previous publications (3–4 kb/min (*Singh and Padgett, 2009*); ~3 kb/min (*Wada et al., 2009*; *Guo and Price, 2013*); 1~6 kb/min (*Ardehali and Lis, 2009*); 3.6 kbp/min (*Fuchs et al., 2014*); 1.8 kbp/min, (*Jonkers et al., 2014*); ~1.5 kbp/min (*Veloso et al., 2014*)). Other fit parameters obtained from the model, such as Pol II loading rates and frequency of initiation will require additional experimental studies to validate. Nevertheless, these model results suggest that a relatively simple input-output mechanism can account for our experimental observations.

## Interference at Pol II clustering step suppresses then re-induces gene bursting at will in living cells

We sought to test the hypothesis that controlling Pol II cluster dynamics can predictably control the β-actin nascent mRNA output. Unfortunately at this time, very little is known about how transient Pol II clusters form, what modulates their dynamics, and how they wield gene expression control. Previously, we found that flavopiridol, a drug that blocks promoter escape, also stabilizes Pol II clusters (*Figure 6—figure supplement 1A*), suggesting a plausible way by which we could control the clustering process (*Cisse et al., 2013*). Here we build on that knowledge, and we test the effects of a reversible drug, 5,6-dichloro-1-β-D-ribofuranosylbenzimidazole (DRB) (*Singh and Padgett, 2009*) on controlling Pol II clustering (*Figure 6—figure supplement 2*) and thus gene expression output (*Figure 6* and *Figure 6—figure supplement 1*).

The accepted mode of action of DRB is the following: when Pol II is first recruited and loaded on a gene promoter, it forms a pre-initiation complex that goes into promoter proximal pausing. CDK9 the kinase of the positive transcriptional elongation factor (P-TEFb) must then phosphorylate Pol II to enable promoter escape, and entry into productive elongation. DRB (as well as flavopiridol) prevents the ATP-binding in CDK9, and thereby maintains initiating Pol II in the promoter-proximal paused state (*Bensaude, 2011*). Indeed, we observed that DRB induces stable Pol II clusters analogous to those observed after flavopiridol treatment (*Figure 6—figure supplement 1B*). While the effect of flavopiridol is irreversible, DRB treatment is reversible, thus enabling revocable control of the drug effect. The reversible nature of the drug treatment lets us test if the presence of a Pol II cluster can cause the burst increase in transcription output.

We serum-induced cells, and then treated them with either flavopiridol or DRB 10 min after serum stimulation. We expect that blocking Pol II-loading with one paused polymerase near the promoter by drug treatment will lead to a gradual decrease in nascent mRNA count. This is because previously loaded and elongating Pol II will complete transcription where as no new Pol II can be loaded. On average the cells will exhibit a gradual decrease in the intensity of MS2 β-actin signal (*Figure 6A,B*). We indeed observe this suppression in the β-actin gene expression profile with both DRB or flavopiridol treatment (*Figure 6C* and *Figure 6—figure supplement 1C*).

Minutes after the β-actin signal disappeared, drug-treated cells featured Pol II clustering exactly at the position where we observed β-actin gene loci (*Figure 6—figure supplement 1D,E*). The observation that Pol II clusters exist at the gene locus, in the absence mRNA synthesis after DRB or

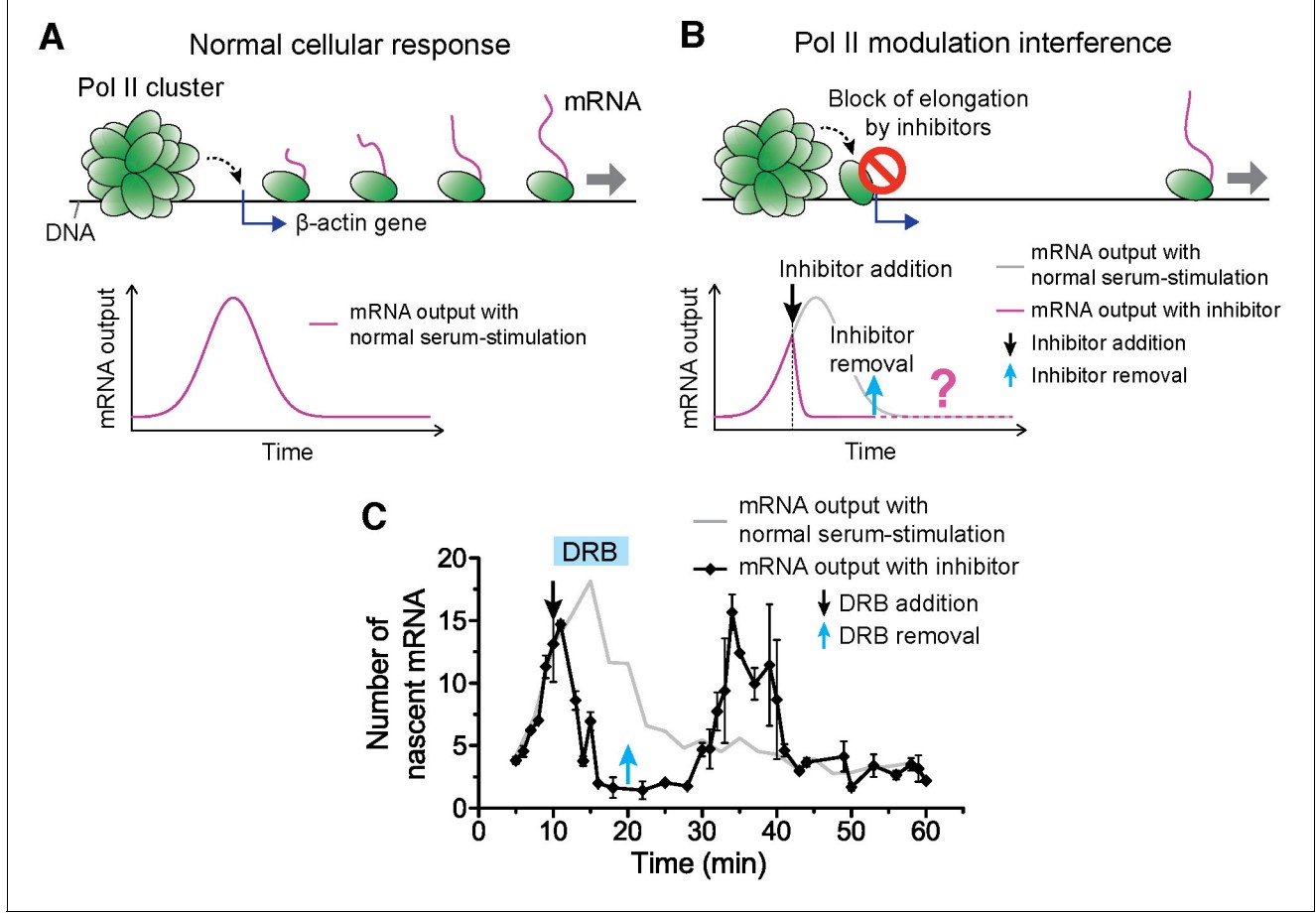

**Figure 6.** Drug interfering with promoter escape and Pol II clustering suppresses then re-induces gene bursting, at will, in living cells. (**A**) With serum stimulation, a stereotyped mRNA response with a single peak can be expected. (**B**) Adding DRB, a drug that prevents promoter escape blocks one Pol II in the promoter proximal paused state thereby preventing new Pol II loading. A gradual decrease in nascent mRNA count is expected without new Pol II loading. Incidentally, we observe that blocking promoter escape leads to stable Pol II on the gene locus and throughout the living cell nucleus (**Figure 6—figure supplement 1E**). According to our model, the presence of this cluster should result in multiple Pol II loading (and thus gene bursting) when the blocking effect of the drug is reversed. (**C**) Data for the mRNA output in the drug treated cells (black) show a decrease in the β-actin signal upon addition of DRB. When the drug is removed, a second burst peak is observed in the mRNA output before a relaxation back to the basal mRNA level. Missing in this assay is the time between escape of the blocking Pol II and the subsequence cluster disassembly. The position of the second mRNA Peak is dependent on when DRB is removed (**Figure 6—figure supplement 1F–I**). The stereotypical β-actin mRNA output response, without DRB treatment, is represented in gray for comparison. For each data point we averaged nascent mRNA counts from a minimum of N=2 living cells imaged at the indicated time after stimulation. Error bars represent standard error of the mean.

The following source data and figure supplements are available for figure 6:

**Source data 1.** Source data for 6C and 6-figure supplement 1F-I.

**Figure supplement 1.** Example time traces of Dendra2-Pol II in flavopiridol and DRB treated cells, transcriptional output response to flavopiridol, co-localization examples of stable Pol II clusters captured at β-actin gene loci minutes after flavopiridol and DRB treatment, and induction of transcriptional bursting at will with DRB.

**Figure supplement 2.** Global effects of DRB and DRB removal in stabilizing Pol II Clusters in living.

flavopiridol treatments, validates our earlier conclusion that Pol II clusters are distinct from elongating polymerase.

We note that the Pol II clusters on the β-actin gene loci, after drug treatment, show evidence of stable clusters, with cumulant graphs featuring a slope onset from the beginning of acquisition

followed by a more gradual plateau (*Figure 6—figure supplement 1D,E*). This observation suggests that drug treatment stabilized Pol II clustering, and is consistent with our previous data on Pol II clusters after drug treatment in human cells (*Cisse et al., 2013*). Since Pol II clusters exist in the absence of actively synthesized β-actin mRNA, this observation corroborates further our conclusion that Pol II clusters do not represent elongating polymerases on the β-actin gene. Rather, these results suggest Pol II clusters formed in a stage of transcription prior to the release of paused pre-initiation complex, a view that is consistent with a lag time between clustering and nascent mRNA detections as observed in *Figure 5*.

We then washed the DRB-treated cells with drug-free medium, so as to release the paused polymerases, and enable new Pol II loading events. The qualitative prediction from our input-output model is that the mere presence of a Pol II cluster will result in the loading of multiple Pol II molecules, and subsequently multiple mRNAs will be synthesized in a burst after drug removal. Consistent with this prediction, we observe a second burst increase in β-actin mRNA synthesis after DRB removal (*Figure 6C* and *Figure 6—source data 1*).

We find that this second burst of transcripts peaks at a time strictly dependent on when we wash DRB (*Figure 6—figure supplement 1F–I* and *Figure 6—source data 1*); thus gene bursting is induced at will in a manner that correlates with the presence of a Pol II cluster at the gene locus. In addition, the second burst appears long after the normal cellular response to serum was expected to return to the basal mRNA count (*Figure 6—figure supplement 1F–I*). This observation implies that Pol II clustering can drive or override the cell's stereotypical response.

The time between the first pause-release and the disassembly of the Pol II cluster, a relevant time-scale in the drug assays, is not accessible in our experiments, limiting further quantitative interpretations. Nonetheless, these results corroborate our findings that the control of cluster dynamics wields a predictable control of a gene's transcriptional output in live mammalian cells.

## Discussion

We captured transient Pol II clusters at the loci of actively transcribed β-actin gene in live MEFs. Our data suggest a transcription regulation mechanism whereby the number of nascent mRNA synthesized is directly correlated with the lifetime of Pol II clusters. The clusters normally occur on the active gene locus during transcription initiation, with a basal average lifetime of ~8 s for β-actin, giving rise to a basal mRNA loading level of ~4 nascent transcripts per allele. Upon stimulation, only the lifetime (and not the size nor the frequency) of Pol II clusters is increased. The longer lifetime during initiation gives rise to a proportional increase in the number of Pol II molecules loaded onto the gene.

A theoretical model recapitulates our observed correlation between Pol II cluster lifetime and mRNA output. The model suggests that during a clustering event, multiple Pol II molecules are loaded on the gene at a rate of one productive Pol II every 2.5 s. With an estimated elongation rate of ~3 kb/min, this loading rate translates into a Pol II packing density of one Pol II complex every ~130 bp, a value close to the maximum density set by the holoenzyme footprint (*Perry et al., 2010*; *Kwak and Lis, 2013*; *Fazal et al., 2015*).

How Pol II clustering is induced, and what controls the dynamics of clustering is currently unknown. We speculate that factors involved in the Pol II pre-initiation complex assembly as well as post-translational modifications that control Pol II promoter escape may also help initiate and control Pol II clustering. However, experimental characterizations are needed to unveil the molecular mechanisms behind Pol II clustering. Nevertheless, our drug treatment results suggest that such detailed mechanistic understanding of how to effect and control Pol II clustering in vivo will provide new ways by which an experimenter may predictably control gene expression, at will.

Our observations support the notion that spatiotemporal clustering of enzymatic factors, through the transient increase of local densities in vivo, are able to effect maximum enzymatic efficiency in a locus-specific manner and when needed. Indeed, beyond components of the Pol II machinery, we speculate that clustering might constitute a general mechanism to build high concentration of factors at discrete genomic loci. For instance, it was recently observed that the nuclear distribution of the pluripotency regulator, Sox2, is highly heterogeneous forming dozens of spatial clusters in the nucleus of embryonic stem cells (*Liu et al., 2014*), and in another study spatial clusters of CDK9 were observed (*Ghamari et al., 2013*). Our findings on Pol II and mRNA output suggest such a local

enrichment of transcription factors can result in a more efficient transcription process in vivo. Because DNA super-coiling has been suggested to play a role in gene bursting in bacteria (*Chong et al., 2014*), it is an intriguing possibility that topological constraint on the DNA may also play a role in cluster-dependent transcriptional regulation.

The combination of our quantitative single cell imaging approach with genome-wide analyses can help uncover the hidden mechanisms behind clustering dependent gene expression regulation. For instance, it was recently reported that docked Pol II molecules accumulate upstream of transcription start sites during starvation in C. elegans, primarily near rapidly responding growth and developmental genes (*Maxwell et al., 2014*) such as the β-actin gene we currently investigated in MEFs. Promoter proximal pausing has also been observed in a wide range of genes (*Kim et al., 2005*; *Muse et al., 2007*; *Zeitlinger et al., 2007*). Our results hint at possible mechanistic relationships between Pol II cluster disassembly and the release of paused polymerase. Although most of beta-actin regulation is thought to be regulated by promoter-proximal sequences (*Danilition et al., 1991*), another important question is whether distal enhancer elements could regulate the initiation and/or lifetime of Pol II clustering by looping to contact the promoter of their target genes. Future studies bridging quantitative single cell imaging of Pol II recruitment with genome-wide mapping of Pol II occupancy and long-distance regulatory interactions will provide important insights in the relationships between chromatin architecture, docking, pausing/release, and Pol II clustering.

The live-cell single-molecule and super-resolution approaches developed in this study can be applied broadly to study other molecular processes in vivo. We emphasize that the multicolor single–molecule and super-resolution approach is general in nature: the approach can be applied, in principle, to any pair of interacting factors that can be labeled. We anticipate that our method will be an important tool to uncover the spatiotemporal organization of the genome in the cell nucleus. While there has been important recent progress in the understanding of chromatin architecture thanks to 3C-related ensemble techniques (*Dekker et al., 2013*; *de Wit and de Laat, 2012*) in fixed cells, our imaging-based single-cell approach provides a powerful assay to address dynamics, and functional output directly in living cells. With the advent of new genetic engineering techniques, molecular enzymes as well as DNA (*Chen et al., 2013*) can be specifically labeled with photo-convertible fluorescent tags, to achieve single molecule studies with high spatial and temporal resolution in living cells.

A major limitation currently is the lack of long-lived fluorescence tags that span the orders of magnitude timescale difference between protein clustering activity and genome response. Here, Pol II clusters lasted just a few seconds, but the transcriptional response occurs on the timescale of minutes. The ability to synchronize a population of cells, with serum starvation, was necessary for the dual-color quantitative super-resolution approach to bridge the different timescales and uncover the correlation between cluster lifetime and mRNA output. The development of new fluorescent molecules adapted for living cells will help improve correlative quantitative imaging of single living cells.

## Materials and methods

### Plasmids

Starting from our original pHAGE-Ubc-NLS-MCP-YFP third generation lentiviral vector (*Lionnet et al., 2011a*), we swapped the YFP with the HaloTag open reading frame (*DeJesus-Hernandez et al., 2011*) to generate the pHAGE-Ubc-NLS-MCP-HaloTag. To achieve this, a HaloTag ORF fragment was generated by PCR and inserted into the digested pHAGE-Ubc-NLS-MCP-YFP vector by ligation. We used the pHAGE-Ubc-NLS-MCP-HaloTag plasmid to create recombinant lentiviral particles (*Mostoslavsky et al., 2006*) generating expression of NLS-MCP-HaloTag driven from the human ubiquitin C promoter in target cells.

The alpha-Amanitin-resistant RPB1 fused to Dendra2 (Dendra2-RPB1Amr) ORF was excised from our original Dendra2-RPB1Amr expressing vector using HpaI and NheI (*Cisse et al., 2013*). We digested the PB53x EF1 Series Piggybac vector (System Biosciences, Palo Alto, CA) with EcoRI, generated blunt ends and performed a second digestion with NheI. The ORF insert was then ligated into the vector using the TaKaRa DNA ligation kit LONG (TaKaRa Bio-Clontech, Shiga, Japan) following the blunt end ligation protocol. Constructs were transformed into Stbl2 competent cells (Life

Technologies, Carlsbad, CA), resistant colonies were then screened by restriction analysis and confirmed by sequencing.

## Generating actb-MBS cell line stably expressing NLS-MCP-HaloTag and Dendra2-RPB1

We used a previously engineered mouse embryonic fibroblasts (MEF) cell line in which a cassette containing 24× MS2 binding sites (MBS) is knocked in the 3′ untranslated region (UTR) of the endogenous β-actin gene (Actb-MBS cells) (*Lionnet et al., 2011a*). In order to generate stable expression of the MS2 capsid protein (MCP) fused to a HaloTag moiety, we incubated the Actb-MBS cells with lentiviral particles generated using the pHAGE-Ubc-NLS-MCP-HaloTag vector. Five days after infection, we stained the cells with the Janelia Fluor 549 (JF549) HaloTag ligand (*Grimm et al., 2015*) by incubating them 15 min at 37°C in growth medium (DMEM, 10% FBS) supplemented with 100 nM fluorescent ligand, and then washed the unbound ligand (15 min in fresh growth medium at 37°C, followed by 2 washes in fresh growth medium). Immediately after staining, JF549-positive cells were sorted using flow cytometry.

In order to generate stable expression of the Dendra2-RPB1Amr construct in our cell line, Actb-MBS cells stably expressing NLS-MCP-HaloTag were transfected with the piggyback vector (PB53x EF1-Dendra2-RPB1Amr) along with a plasmid expressing the super Piggybac Transposase using an Amaxa nucleofector (Lonza, Basel, Switzerland). Cells expressing Dendra2-RPB1-Amr were then selected with alpha-Amanitin (2 ug/mL, Sigma-Aldrich, St. Louis, MO) for 2 weeks, starting 2 days post-transfection (*Cisse et al., 2013*). After the 2-week drug-selection period, cells were stained with the Janelia Fluor 646 ($JF_{646}$) ligand (*Grimm et al., 2015*), using the same protocol as $JF_{549}$ (see above). Double positive cells (green fluorescence from Dendra2; red fluorescence from $JF_{646}$) were sorted using flow cytometry. Cells were derived in-house from primary MEFs. Following immortalization and integration of fluorescent labels, cell identity was regularly controlled by visualizing the fluorescence in the Dendra2 and MCP-HaltoTaq channels (assaying correct nuclear localization for both proteins, photo-conversion upon 405 excitation for Dendra2, and presence of hundreds of mRNA particles in the MCP-HaloTaq channel). The cell lines undergo regular mycoplasma contamination testing by the Janelia Cell Culture Facility and also at the Massachusetts Institute of Technology.

## Cell culture

The cells were cultured and maintained in DME (Dulbecco's Modified Eagle's) Glutamax media (10567) from Thermo Fisher Scientific (Cambridge, MA) supplemented with 10% FBS (Fetal Bovine Serum) (26140, qualified, US origin) from Gibco and penicillin/streptomycin (10 U/ml penicillin and 10 μg/ml streptomycin) (15140) from Gibco. The cells were grown in 37°C incubator containing 5% $CO_2$ in a water-saturated atmosphere.

## Serum starvation and serum induction

Cells were plated on 25 mm round glass coverslips (CS-25R) from Warner Instruments (Hamden, CT) and were first grown in DME media containing 10% FBS. The cells were then transferred to serum-free media (DME media and penicillin/streptomycin) after a confluency of 70% was reached. The cells were maintained in serum-free media overnight before live cell imaging. For serum-induction experiments the serum-starved cells were treated with 10% FBS in L-15 (Leibovitz) medium, prior to imaging.

## Flavopiridol and DRB treatment

The cells were plated on 25 mm round glass coverslips (CS-25R) from Warner Instruments and maintained in DME media containing 10% FBS and penicillin/streptomycin until a confluency of 70% was reached. The cells were treated with 10 μM flavopiridol hydrochloride hydrate (F3055, from Sigma-Aldrich) for flavopiridol inhibition, and 100 μM DRB (5,6-Dichlorobenzimidazole 1-β-D-ribofuranoside, D1916, from Sigma-Aldrich) for DRB inhibition prior to live-cell image acquisition. At these concentrations drug effects can be observed within minutes in individual cells, compared to hours in lower concentration. For DRB removal experiments, DRB supplemented medium was exchanged with 10% FBS in L-15 (Leibovitz) medium.

## Cell fixation

Cells grown in DMEM containing 10% FBS and penicillin/streptomycin, were washed with $1\times$ PBS and fixed in 4% Paraformaldehyde (15714) from Electron Microscopy Sciences (Hatfield, PA 19440) for 15 min at room temperature. This was followed by washing with $1\times$ PBS three times. For each washing step we incubated cells in PBS for approximately 2 min to ensure efficient removal of formaldehyde.

## Control experiment with Dendra2 alone

For imaging of Dendra2 alone, we transiently transfected wild-type MEFs with pDendra2-C plasmid (632546) from Clontech (TaKaRa Bio-Clontech). Transfection was carried out using X-tremeGENE 9 DNA Transfection Reagent (06365787001) from Roche (Basel, Switzerland), according to the manufacturer's instructions. Following transfection, cells were transferred to DME media containing 10% FBS and penicillin/streptomycin and grown at 37°C with 5% CO2 for at least 24 hr before imaging.

## Incubation of JF646 for imaging β-Actin mRNA

The cells were plated on 25 mm round glass coverslips and grown until a confluency of 70% was reached. The cells were then transferred to serum-free media (DME media and penicillin/streptomycin) and maintained in the serum-free media overnight before addition of HaloTag ligand conjugated Janelia Fluor (JF646-HaloTaq ligand). For live-cell labeling we incubated cells with 500 nM JF646-HaloTag ligand for 1 hr in the serum-free media followed by washing with 1x PBS. The cells were then transferred to L-15 media for imaging.

## Super-resolution (PALM and STORM) imaging

Super-resolution imaging was performed using a Nikon Eclipse Ti microscope with a $100\times$ oil immersion objective (NA 1.40) (Nikon, Tokyo, Japan). Activation (405 nm for conversion of Dendra2) and excitation (488 nm for pre-converted Dendra2, 561 nm for post-converted Dendra2, and 642 nm for JF646) laser beams were combined in an external platform; the combined beam was expanded and re-collimated with an achromatic beam expander (AC254-040-A and AC508-300-A, from THORLABS, Newton, NJ), and focused with an achromatic converging lens (#45–354, from Edmund Optics, Barrington, NJ) into the rear plane of the objective. Images were acquired with an Andor iXon Ultra 897 EMCCD camera using Micro Manager 1.4, a free and open-source software (Andor Technology, Belfast, United Kingdom) (*Edelstein et al., 2014*).

For PALM imaging of Dendra2-RPB1 expressing cells, DME media was substituted with L-15 media, without phenol red (21083) from Thermo Fisher Scientific, supplemented with 10% FBS (Fetal Bovine Serum) (26140, qualified, US origin) from Thermo Fisher Scientific. The cells were maintained at 37°C in a temperature controlled platform (InVivo Scientific, St. Louis, MO) on the microscope stage during image acquisition. Image sequences were acquired at a rate of 60 ms per frame under illumination with 405 nm for photo-conversion and 561 nm for excitation, both with the EM-gain 900 on Andor iXon Ultra 897 EMCCD. Z-position of the microscope stage was maintained during acquisition using the Perfect Focus System (PFS) of the Nikon Ti Eclipse. The laser power densities used for Dendra2 were 1.3 W/cm2 (405 nm) and 3.2 kW/cm2 (561 nm) on the image plane.

For STORM imaging of JF646, the live cell sample was incubated with 500 nM JF646 for 1 hr. The cells were transferred from DME media into L-15 media containing 10% FBS before image acquisition. Images were acquired at 60 ms time-resolution with illumination of a 642 nm laser at a power density of 2.5 kW/cm2. Images were acquired on the same EMCCD camera (Andor iXon Ultra 897 EMCCD) with a reduced EM-gain of 400 to avoid saturation. Z-stack images were acquired by moving the microscope stage from a plane selected at the bottom of cell nucleus set as zero position (0 μm) up to 5 μm, with a step size of 200 nm, using the multi-dimensional acquisition module of Micro Manager 1.4 (*Edelstein et al., 2014*).

## Nascent β-Actin mRNA estimation

The number of nascent mRNA at the bright β-actin focus was estimated by comparing the JF646 intensity at the focus with the intensity of diffusing single mRNA molecules. Each single cell was imaged for 100 frames at 20 frames per second with 642 nm laser illumination at a power density of 2.5 kW/cm2. The images were then merged by maximum intensity projection using ImageJ.

Intensities of single spots were measured by summing the intensity values within a $3 \times 3$ pixels window, then subtracting off the average background noise intensity of the cell nucleus. The intensity of each focus was divided by the average intensity of single mRNA molecules to calculate the number of mRNA molecule at the actively transcribing bright foci.

## Image analysis and super-resolved images

Analysis of super-resolution image was performed as described before (*Izeddin et al., 2013*; *Cisse et al., 2013*). Raw image sequence was analyzed with a custom adaptation of the multiple-target tracking algorithm (MTT) (*Serge et al., 2008*). For localization of detected fluorophores, the point-spread function (PSF) of spatially separated individual single fluorescence was fitted with a 2-D Gaussian distribution. The center of the Gaussian fit was used as a precise estimate of the position of the fluorophore. The live cell localization precision was measured to be 31 nm for Dendra2 and 18 nm for JF646 (Figure S2D–S2I). Regions of interest were selected from a pointillist reconstruction of the localizations, and temporal clusters within the regions of interest were identified using our custom analysis software, available on the Cissé lab GitHub account (https://github.com/cisselab).

## Single molecule fluorescence in-situ hybridization method

For smFISH quantification, we followed a protocol detailed in (*Lionnet et al., 2011a*) and summarized as follows: Immortalized MEF cells were seeded onto glass coverslips (50,000 cells onto an 18 mm diameter #1.5 round coverslip), and fixed 24 hr later in 4% paraformaldehyde (10 min, room temperature). After fixation, cells were rinsed in PBS and permeabilized 10 min in PBS supplemented with 0.5% Triton X-100. After rinsing, cells were incubated in 2xSSC + 10% Formamide for 10 min. Hybridization was performed over 3 hr at 37 degrees, in 2xSSC, 10% formamide, 2 ng/uL probe mix, 10% Dextran Sulfate, 2 mg/mL BSA, 0.05 mg/mL *E coli* tRNA, 0.05 mg/mL salmon sperm ssDNA. The probe mix consistend in 3 probes targeting the MBS cassettes, plus 36 probes targeting the beta-actin mRNA (probe sequences in *Lionnet et al., 2011a*). All probes were carrying a single Cy3 label. After hybridizations, cells were washed twice 20 min in 2xSSC+10% formamide at 37degrees, then counterstained with DAPI, rinsed in PBS and mounted onto a slide using the Prolong Gold mounting medium (Life Technologies).

## Monte carlo simulation

Monte Carlo simulations of tcPALM based on Dendra2 photophysical dynamics were written in Matlab, (the custom m-file code available on GitHub at https://github.com/timotheelionnet/Pol-II-Clustering-Photophysics). Clusters were simulated as 100 Pol II molecules assembling and disassembling at times set by the user (e.g. we modeled dynamic clusters as 100 molecules assembling at 200s and disassembling at 210 s; static clusters were modeled as clusters assembling at 200 s without disassembly). While the cluster was present, each of the 100 molecules trajectories was independently modeled using the Gillespie algorithm, with the molecular states consistent with our photophysical model (Appendix 4–5, *Figure 3—figure supplement 1*). The molecules were allowed to turn on ($k_{on}$ = 0.0067 s$^{-1}$); once on, they could either blink to a dark state ($k_{dark}$ = 9.6 s$^{-1}$) or photobleach ($k_{bleach}$ = 3 s$^{-1}$). Blinking molecules could turn back to the on state with a rate $k_{rev}$ = 2.33 s$^{-1}$. All transition rates were set to our experimentally measured values. Once the time of all molecular transitions was determined, trajectories were binned with a 60 ms step to mimic camera acquisition. Simultaneous detections of two or more molecules were only counted as a single count to reflect our data analysis which does not take the spot intensity into account. Blinking events shorter than half a frame were left undetected to model our limited camera sensitivity.

## Acknowledgements

We thank Jeff Gore (MIT), Arjun Narayanan (MIT), Jeremy England (MIT), Sebastian Lourido (MIT), Hervé Rouault (Janelia Research Campus) and Arjun Raj (UPenn) for helpful comments. K McGowan, M Ramirez and X Zhang (Janelia Molecular Biology Shared Resource) provided assistance in cloning; H White and Z Ma (Janelia Molecular and Cell Culture Resource) assisted with cell culture and Flow Cytometry sorting. Research reported in this publication was supported by the National Cancer Institute of the National Institutes of Health under Award Number DP2CA195769 to IIC. The content is solely the responsibility of the authors and does not necessarily represent the official views of the

National Institutes of Health. This work was also supported by funds from the MIT Department of Physics and the Howard Hughes Medical Institute.

## Additional information

### Funding

| Funder | Grant reference number | Author |
|---|---|---|
| National Institutes of Health | NIH Director's New Innovator Award (DP2CA195769) | Ibrahim I Cisse |
| Howard Hughes Medical Institute | Transcription Imaging Consortium | Timothée Lionnet |

The funders had no role in study design, data collection and interpretation, or the decision to submit the work for publication.

### Author contributions

W-KC, Acquired and analyzed all data in the main manuscript, Conceived the time dependent mRNA output and cluster lifetime experiments, Conceived the drug experiments, Drafting or revising the article, Contributed unpublished essential data or reagents; NJ, Acquisition of data, Analysis and interpretation of data, Drafting or revising the article, Contributed unpublished essential data or reagents; BPE, Conceived the project, Conceived and acquired H2B control data, Analysis and interpretation of data, Drafting or revising the article, Contributed unpublished essential data or reagents; TI, Co-built the original microscopes used for imaging, Acquisition of data, Contributed unpublished essential data or reagents; JOA, Developed the analyses software used, Co-developed the molecular counting analysis, Acquisition of data, Analysis and interpretation of data, Drafting or revising the article, Contributed unpublished essential data or reagents; WC, Acquisition of data, Analysis and interpretation of data, Contributed unpublished essential data or reagents; JBG, LDL, Developed the JF dyes, Contributed unpublished essential data or reagents; J-HS, Co-developed the molecular counting analyses, Analysis and interpretation of data, Drafting or revising the article; TL, Conceived the project, Supervised all aspects of the cell engineering and the theoretical modeling, Analysis and interpretation of data, Drafting or revising the article; IIC, Supervised all aspects of the project, Conceived the project, Analysis and interpretation of data, Drafting or revising the article

### Author ORCIDs

Won-Ki Cho, http://orcid.org/0000-0001-9336-0686
Ibrahim I Cisse, http://orcid.org/0000-0002-8764-1809

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

# Appendix 1: Pair-correlation PALM (pcPALM) analysis

The two-dimensional pair correlation function g(r) represents the probability of particle detection at distance r from a given detected point in a two dimensional imaging plane (*Sengupta et al., 2011*). The implementation was previously described (*Cisse et al., 2013*).

Briefly, if the distribution of proteins is purely random the pair correlation g(r) is given:

$$g(r) = g^{PSF}/\rho + 1 \tag{1}$$

with

$$g^{PSF} = \left(1/4\pi\sigma^2\right) \times \exp\left(-r^2/4\sigma^2\right)$$

where $\rho$ represents the average surface density of molecules, $g^{PSF}$ denotes the correlation of uncertainty in the point spread function (PSF). $\sigma$ denotes standard deviation of distribution of detections of single protein. All data were first subjected to this distribution as a null hypothesis.

In the case where the null hypothesis is rejected, and protein correlation may be approximated by $g^{protein}(r) = A \times \exp(-r/\xi) + 1$. Thus the total correlation will be given by,

$$\begin{aligned} g(r) &= g^{stoch}(r) + g^{protein}(r) * g^{PSF}(r) \\ &= g^{PSF}/\rho + (A \times \exp(-r/\xi) + 1) * g^{PSF}(r) \end{aligned} \tag{2}$$

where $\xi$ denote the correlation length of protein cluster, and $A$ denote the amplitude of protein correlation. $*$ is the convolution operator.

For pcPALM analysis, the pair-correlations of Dendra2-alone (fixed cell) and Dendra2-Pol II (fixed cell and live cell) were first subjected to *Equation 1*. Data of Dendra2-alone in fixed cells were well fitted by *Equation 1* while Dendra2-Pol II data in both fixed and live cells did not fit well with *Equation 1*, indicating that Dendra2-alone molecules are distributed randomly in cell nucleus, whereas Dendra2-Pol II are not. The data were then fitted with *Equation 2*, Dendra2-alone data were mainly fitted by the stochastic component of *Equation 2* and the amplitude (A) of protein correlation and the correlation length ($\xi$) of it were negligible, whereas the protein correlations for Dendra2-Pol II were apparently influenced to the fitting of *Equation 2*, more effectively in live cells (*Figure 1—figure supplement 1A–C*).

The pointillist data were obtained by 2D-Gaussian fitting of the point-spread function for individual particles using the MTT software package and the two-dimensional correlation function was computed using the 2D Fast Fourier Transform (fft2) function in MATLAB as described previously (*Sengupta et al., 2011*). Fitting of correlation data was performed using Curve Fitting Toolbox (cftool) in MATLAB.

## Appendix 2: Pol II vs H2B single particle tracking PALM (sptPALM) image acquisition and analysis

Data in *Figure 1—figure supplement 1F,G* were independently recorded under identical imaging condition on a custom-built 3-camera RAMM frame microscope (ASI, Eugene, OR) using an Olympus 1.4 NA PLAPON 60x OSC objective (Olympus, Tokyo, Japan), and a custom tube lens (LAO-300.0, Melles Griot, Albuquerque, NM), resulting in 100x overall magnification (*Grimm et al., 2015*; *Piatkevich et al., 2014*). A 2 mm thick quad-band excitation dichroic (ZT405/488/561/640rpc, Chroma, Cambridge, MA), a 2 mm thick emission dichroic (T560lpxr, Chroma), and a band-pass emission filter (FF01-609/54-25, Semrock, Rochester, NY) filtered the emitted light. Stroboscopic 405 nm excitation of the Stradus 405–100 laser (Vortran, Sacramento, CA) was achieved using a NI-DAQ-USB-6363 acquisition board (National Instruments, Austin, TX), which also controlled the 555 nm laser (CrystaLaser, Reno, NV). Z-stacks through the entire nucleus were generated by stepping through a sample using four 0.6 µm steps using a Piezo scanning stage (P-454.3C7, Physik Instrumente, Auburn, MA). Fluorescence images were recorded with a back-illuminated EMCCD camera (Andor Technology, Ixon Ultra DU-897U, 17 MHz EM amplifier, 300 x gain, center quadrant) at 20 frames/s. Dendra2 was photo-converted by 0.5 ms long excitation pulses of 405 nm (10 W/cm$^2$) every 2000 ms. During all imaging experiments, cells were maintained at 37°C in a humidified 5% CO2 3% O2 and 92% N2 environment supplied by a live-cell incubator (Tokai Hit, Shizuoka-ken, Japan).

We detected individual molecules and measured their 2D position using a maximum intensity projection of the stacks generated at each time point, using a MATLAB software previously described (*Lionnet et al., 2011a*). We manually generated mask images encompassing the cell nucleus in each movie using ImageJ. The nucleus was subdivided in a square grid (the sides of each square within the grid were set to 500 nm), and the molecule detections were assigned to the corresponding unit cell of the grid based on their position. Grid cells with more than 10 detections over the duration of the movie were automatically sorted. Time traces of cumulated detections per cell were then randomly selected from the set of cells with more than 10 detections and plotted. All analysis steps were performed using MATLAB code.

## Appendix 3: Precision of localization measurement

The localization precision of individual fluorescent molecules was investigated for our live-cell super-resolution microscope using the approach described by Thompson, Larson and Webb (**Thompson et al., 2002**) and adapted for multi-color PALM (**Betzig et al., 2006**). In this approach the precision of localization is given by,

$$\left\langle (\Delta x)^2 \right\rangle = \frac{s^2 + a^2/12}{N} + \frac{4\sqrt{\pi}\, s^3 b^2}{a N^2}$$

where $s$ is standard deviation of the PSF, $a$ is the size of pixel, $N$ is total number of photons of a signal and $b$ is number of the back ground photons (**Thompson et al., 2002**; **Betzig et al., 2006**; **Shroff et al., 2007**).

To measure the detection accuracies for Dendra2 and JF646 in this study, we determined the total number of photons of fluorophore and the background noise from the intensity profile of detections, considering the EM-gain, electron amplification level and quantum efficiency of Andor Ultra iXon 897 EMCCD for each wavelength. The standard deviations were obtained by 2D-fitting of the PSF, and the pixel size was 160 nm for imaging with 100X magnification objective lens on the CCD panel of original pixel size of 16μm. The results for dual-color super-resolution are depicted in **Figure 2—figure supplement 1D–I**.

# Appendix 4: Photo-physical simulation of Dendra2 fluorescence emission

MEF cells were fixed using paraformaldehyde at 12 min after serum stimulation. These samples were then imaged to exhaustion. Putative beta-actin transcription loci were chosen by selecting the brightest foci in a summed-intensity projection of the cells. Regions of interest (ROIs) were selected by identifying the grouping of polymerase localizations that was closest to the beta-actin locus.

To obtain estimates for the number of molecules within a region of interest, the fluorescence emission of the photo-convertible fluorescent protein Dendra2 was modeled as a four state system (*Figure 3—figure supplement 1A*). Upon 405-nm irradiation, Dendra2 is switched from the pre-converted, green fluorescent state to the converted, red fluorescent state.

From the red fluorescent state, the molecule can transition to a transient, reversible dark state with rate $k_{dark}$ = 9.61/s or an irreversibly bleached state with rate $k_{bleach}$= 3.0/s. Recovery from the dark state with rate $k_{rec}$ = 2.33/s leads to blinking of the fluorophore. The average number of blinking cycles is given by the expression, $k_{dark}/k_{bleach}$ = 3.2. All rate constants were determined from fixed cell controls under conditions identical to the actual experiments (*Figure 3—figure supplement 1B–D* and see Appendix 5: Estimation Photo-physical Parameters of Dendra2).

We used two processes to determine the number of molecules within a cluster. First, the number of molecules in each ROI was estimated using an Aggregated Markov Model approach, as described in *Rollins et al.* (***Rollins et al., 2015***). For each region, the likelihood of a given trajectory was calculated for varying guesses of the number of molecules, and the number which maximized the likelihood of the trajectory was called the estimate for the number of observed molecules. As can be seen in *Figure 3D*, the ROIs can be roughly partitioned into two groups on the basis of the cumulative number of localizations. These two groups are tentatively identified as 'Clustered' and 'Un-clustered'. We were able to obtain estimates for the number of molecules inside of the 'Un-clustered' ROIs. This number ranged between 1 and 20, consistent with the expected number of elongating polymerases. 'Clustered' regions, however, contain a much greater number of molecules. In these cases, the running time for the software was too long for accurate estimates to be obtained for the number of molecules in a cluster. We were, however, able to find a lower bound of approximately 60 molecules.

To obtain estimates for the number of molecules within the 'Clustered' regions, we made use of a second approach using single molecule simulations. To simulate fluorescence emission from a cluster of a number of molecules $N_{sim}$, the time-course from photo-conversion to irreversible bleaching was calculated for each of the molecule. Subsequently, the continuous time simulation was binned according to the kinetic cycle time of 60 ms. When any of the molecules resided in the red fluorescent on state for more than half of the frame interval, a detection was counted in the respective frame. Note that multiple molecules in the on state at the same time can only lead to a single detection. The cumulant of all detections in a cluster was calculated for comparison with experimental data. Wait times for photo-activation were drawn randomly from a bimodal exponential distribution with rate constants determined by fitting the experimentally observed cumulant with the function,

$$C(t) = N \cdot \left(1 - a \cdot \exp\left(-k_{on,1} \cdot t\right) - (1 - a) \cdot \exp\left(-k_{on,2} \cdot t\right)\right).$$

The number of blinks was drawn from a geometric distribution with an average of 3.2. The dwell times in the visible and transient dark states were drawn from an exponential distribution with times constants of 79.8 ms and 431 ms respectively (*Figure 3—figure supplement 1B–D*).

To estimate the number of molecules in a cluster, the experimentally observed cumulant was compared to results for 100 simulated clusters each for values of $N_{sim}$ = 1–200. The mean squared deviation between experimental and simulated cumulant was calculated and averaged over all 100 simulations. The value of $N_{sim}$ yielding the minimum average deviation is stated as the most likely number of molecules in the respective cluster (*Figure 3—figure supplement 1E*).

We note that this approach is less precise than Aggregated Markov Model approach. However, it yields similar results for cumulants from the 'Un-clustered' regions, but in a fraction of the time. This allowed us to obtain the estimates for clusters that were too large for the aggregated markov model approach.

## Appendix 5: Estimation of photo-physical parameters of Dendra2

In order to determine that photo-physical parameters of the Dendra2 molecule, we performed experiments where we transiently transfected U2OS cells with free Dendra2, and then fixed the cells with paraformaldehyde. Free Dendra2 was chosen for the experiment to ensure that the fluorophores would not cluster on small length scales, and the cells were fixed to eliminate any biological dynamics. Single molecules were identified from the pointillist reconstruction by exhaustively choosing regions of interest containing more than one localization approximately within a pixel (160 nm). Localizations within a region of interest were then group into independent events. Groups of localizations larger than 2 that occurred within a fixed region of space and were well separated in time from other localizations were identified as single molecules.

Within a single Dendra2, we could count the number of blinking events as the number of continuous stretches of bright frames to generate the blink number distribution. The average lifetime in the bright state and in the blink state were determined by examining the dwell time distribution of stretches of continuous localizations and stretches of a continuous lack of localizations respectively. From the average number of blinks, and the average lifetimes in the bright and blink states, we could calculate three of the four rate constants in our model, $k_{dark}$, $k_{bleach}$, and $k_{rec}$ (*Figure 3—figure supplement 1A*). In total, we identified 2627 proteins. All data is an aggregate of 13 cells imaged over the course of three separate days (*Figure 3—figure supplement 1B–D*).

The fourth rate constant, $k_{on}$ can be measured by fitting the cumulative number of detections in the entire nucleus to an exponential plus a constant background detection rate,

$$C(t) = A \left( 1 - e^{-t/\tau} \right) + B\,t$$

where $C(t)$ is the cumulant, $A$ is the estimated number of localization generated by the fluorophore and $B$ is the constant background detection rate, $1/\tau$ is the empirical activation rate. Because the empirical activation rate ($\sim 1/150$ s$^{-1}$; data not shown) is two orders of magnitude slower than the three previously determined rate constants, we expect that $k_{on}$ dominates the empirical activation rate. Indeed, simulations using our estimated transition rates reveal that the fit time constant of the cumulant is equivalent to the activation time constant, $k_{on}$, of the simulation within 2.6% (Data not shown). We therefore identify the empirical rate constant as $k_{on}$, and can easily estimate it in live cells for each individual cell, and fixed cell experiments are unnecessary for its determination.

Our measured parameters are in fair agreement with the results of S-.H. Lee *et al.* (*Lee et al., 2012*) considering the differences in laser illumination intensities and the different chemical environment of fixed cell nuclei versus in vitro.

# Appendix 6: Theoretical model of Pol II clustering and transcription

After serum stimulation, cells typically displayed a unique transcription burst lasting 15–30 min. We tested whether a model where short Pol II clustering events (seconds), which duration is modulated over time scales of minutes, could explain the nascent transcription temporal profiles we measured in our experiments. The main model assumption was that productive beta-actin transcription initiation could only occur during clustering events. Clustering events occured stochastically as a zeroth order reaction with rate $f_{\text{cluster}}$ ($f_{\text{cluster}}$ remains constant over time to match our experimental observations, **Figure 5—figure supplement 1B**). We assumed that mRNA initiation events could occur at random during each clustering event, with a constant probability per unit time, $f_{\text{mRNAinit}}$. The duration of each cluster was assumed to be exponentially distributed with a time-dependent average duration $\langle \tau(t) \rangle$. From the moment the mRNA initiated, the elongation process was modeled as a deterministic process, with a constant elongation rate, $k_{\text{elong}}$ followed by a termination step with fixed duration, $t_{\text{term}}$ (**Larson et al., 2011**; **Lionnet et al., 2011b**). The fluorescence profile of the nascent mRNA as a function of time after initiation, $MS2(t)$ was modeled the following way: zero before Pol II reaches the MBS cassette; a linear ramp through the cassette as more loops are transcribed; a plateau for the remainder of elongation until termination (defined as the mRNA leaving the transcription site). Based on the model assumptions, one can compute the average nascent chain fluorescence as a function of time $\langle N(t) \rangle$ as follows: the average nascent chain fluorescence at the gene locus is the result of the convolution of the fluorescence timetrace of one nascent mRNA $MS2(t)$ by the frequency of productive initiation events $\langle P_{prod\ inititiation}(t) \rangle$:

$$\langle N(t) \rangle = MS2(t) \otimes \langle P_{prod\ inititiation}(t) \rangle = \int_{u=-\infty}^{t} \langle P_{prod\ inititiation}(u) \rangle\, MS2(t-u)\, du \tag{3}$$

where

$$MS2(t) = 0, \quad t < t_s \overset{\text{def}}{=} \frac{L_{\text{upstream}}}{k_{\text{elong}}}$$
$$MS2(t) = \frac{t-t_s}{t_p-t_s}, \quad t_s \leq t \leq t_p \overset{\text{def}}{=} \frac{L_{\text{array}}+L_{\text{upstream}}}{k_{\text{elong}}}$$
$$MS2(t) = 1, \quad t_p \leq t \leq t_p + \frac{L_{\text{downstream}}}{k_{\text{elong}}} + t_{\text{term}}$$

with the following notations (all lengths in nucleotides):

$L_{\text{upstream}}$ = 3425 nt (length separating the transcription start site from the start of the MBS cassette)

$L_{\text{array}}$ = 1256 nt (length of the MBS cassette)

$L_{\text{downstream}}$ = 309 nt (length separating the end of the MBS cassette from the polyadenylation site)

The frequency of productive initiation events $\langle P_{prod\ inititiation}(t) \rangle$ is the product of the probability to find a cluster at the gene $f_{\text{cluster}} \langle \tau(t) \rangle$ (following from zeroth order kinetics) by the initiation frequency from a cluster ($f_{\text{mRNAinit}}$):

$$\langle N(t) \rangle = \int_{u=-\infty}^{t} \tau(u) f_{\text{cluster}} f_{\text{mRNAinit}}\, MS2(t-u)\, du \tag{4}$$

In **Equation 4**, the time profile $\tau(t)$ is measured (**Figure 5A**); the only unknown variable in $MS2(t)$ is the elongation rate $k_{\text{elong}}$ (we set a the termination time to 1 min in order to account for 3'end processing; this ~1 min delay is consistently observed across a wide variety

of biological systems (*Larson et al., 2011*; *Boireau et al., 2007*; *Zenklusen et al., 2008*; *Brody et al., 2011*)); also unknown are $f_{mRNAinit}$ and $f_{cluster}$ (although we measure an effective value for $f_{cluster}$, the sparse nature of the tcPALM observation technique means that we most likely miss some clusters, i.e. underestimate $f_{cluster}$).

As $f_{cluster}$ and $f_{mRNAinit}$ only appear as a product in our equation, the fitting is underdetermined: given a best fit value for the product $F$, there is an infinite number of ($f_{cluster}\ f_{mRNAinit}$) pairs that satisfy $F = f_{cluster}\ f_{mRNAinit}$. This means that models with identical $F$ values but different pairs of parameters ($f_{cluster}$; $f_{mRNAinit}$) cannot be discriminated. The physical interpretation of this indetermination is the following: models with infrequent clusters (low $f_{cluster}$) can generate the experimentally observed average nascent transcription level provided they rapidly fire mRNAs within each cluster (high $f_{mRNAinit}$); this would correspond to infrequent clusters producing each a large number of mRNAs. Conversely, the same average nascent transcription profile can be obtained with more frequent clusters, each producing a modest number of productive initiation events (high $f_{cluster}$, low $f_{mRNAinit}$).

Models with the same $F$ value but different ($f_{cluster}$ ; $f_{mRNAinit}$) pairs might yield identical average nascent transcription profiles but they differ in the variability between nascent transcription trajectories of single alleles. We used the fact that our experiments have single allele resolution to solve the indetermination stemming from fitting only the average transcription level (*Equation 4*). We measured the variance in the number of nascent chains as a function of time. The theoretical variance of the nascent chain fluorescence as a function of time $\sigma(t)$ can be calculated as previously:

$$\sigma(t) = \left\langle N^2_{\mathrm{nascentchains}} \right\rangle(t) - \left\langle N_{\mathrm{nascentchains}} \right\rangle^2(t)$$

where

$$\left\langle N^2_{\mathrm{nascentchains}} \right\rangle(t) = 2 \iint_{u \leq v = -\infty}^{t} f^2_{\mathrm{mRNAinit}}\, f_{\mathrm{cluster}}\, \tau(u) \left\{ e^{-\frac{v-u}{\tau(u)}} + \left(1 - e^{-\frac{v-u}{\tau(u)}}\right) f_{\mathrm{cluster}}\, \tau(v) \right\}$$
$$MS2(t-u)\, MS2(t-v)\, du\, dv$$

Fits were performed as follows:

1) We approximated the experimental time trace $\tau(t)$ as a Gaussian profile offset by a baseline using a non-linear least squares fit. This step provides a simple formula for $\tau(t)$ that accurately reflects the data while simplifying subsequent calculations.

2) We computed the integral in *Equation 4* numerically using the global adaptive quadrature approximation (Matlab). We wrote custom Matlab software (code available on GitHub at https://github.com/timotheelionnet/Pol-II-Stochastic-Dynamics) to fit the numerical integral $\langle N(t) \rangle$ to our experimental data for the nascent chains (chi-squared minimization using the Quasi-Newton method). The floating parameters were the elongation rate $k_{elong}$ (implicit in the profile $MS2(t)$) and the product $F = f_{cluster}\ f_{mRNAinit}$.

3) Using the values of $k_{elong}$ and $F$ obtained from the fit in 2), we then fitted our measurements for the variability in the number of nascent chains as a function of time to the predicted formula for $\sigma(t)$. Once again, we used a global adaptive quadrature approximation for the integral and fitted the numerical integral $\sigma(t)$ to our experimental data using chi-squared minimization (Quasi-Newton method). The only floating parameter in this fit was $f_{cluster}$.

The values extracted from the fit are the following: $f_{cluster}$= 1.37 min$^{-1}$; $k_{elong}$= 3.1 kbp/min; $f_{mRNAinit}$= 0.43 s$^{-1}$. The value of $f_{cluster}$ is higher than the observed value; this is not surprising as our method tcPALM only allows observing a fraction of all Pol II molecules at each given time, and therefore is subject to missing events (see *Figure 5—figure supplement 2*).

