## [Decision Letter]

Thank you for submitting your work entitled "RNA Polymerase II cluster dynamics predict mRNA output in living cells" for consideration by *eLife*. Your article has been favorably evaluated by Kevin Struhl (Senior editor) and three reviewers, one of whom is a member of our Board of Reviewing Editors.

The following individual involved in review of your submission has agreed to reveal their identity: John Lis (peer reviewer).

The Reviewing Editor and reviewers have discussed the reviews with one another and the Reviewing Editor has drafted this decision to help you prepare a revised submission.

Summary:

The reviews are in general favorable. All three reviewers found the discoveries presented in the manuscript to be very interesting and the methodology used state-of-the-art and elegant. A fairly broad audience should appreciate the findings. The paper is well written.

Essential revisions:

The reviewers raised the following major points that need to be addressed before the manuscript is accepted for publication.

1) The authors showed the presence of two types of Pol II clusters with distinct sizes, one with a substantially larger number of Pol II molecules than the other. They conclude that the smaller clusters represent elongating polymerase molecules on a single gene and the larger clusters present a distinct state, which is the focus of this paper. While these suggestions are reasonable, they lack experimental validation. In particular, there is no experimental evidence convincingly support that the smaller clusters represent elongating polymerase molecules on a single gene. The authors should provide such a proof which we believe could be completed within the two months we allow for return of a revised manuscript. Once this is proven, it will be clear that the clusters with a much larger molecule number represent a state distinct from elongating polymerase.

2) The authors use comparison of the apparent cluster lifetimes between live and fixed cells to conclude that the clusters are dynamic and transient in live cells. This is suggestive but not conclusive, since the photoswitch times in live and fix cells could be different. A more convincing demonstration that the transient cluster lifetime is due to transient cluster dynamics rather than photobleaching of the fluorescent proteins is to measure whether the apparent lifetime is dependent on laser intensity.

3) Results: "We interpret the inflection points with sudden transitions from very high to very low frequency of detections as cluster disassembly events." It is so not precise to define a plateau of a cluster's time trace as a cluster disassembly event. How is a pausing event distinguished from a disassembly event in tcPALM? For the same reason, it is not precise to define the abrupt slope as the cluster lifetime in the case that the plateau is caused by pausing of Pol II recruitment.

4) Comparing Figure 5 and Figure 5—figure supplement 1, why does the rate of detection (clustering strength) remain nearly a constant while Pol II cluster lifetime has a significant increase between 10-15 mins after serum stimulation. What does longer lifetime exactly mean, number of pre-existing Pol II, fast recruitment of Pol II or slow dissociation of Pol II? And why does lifetime increase upon serum stimulation?

5) Results: The authors claimed that "Alternatively, we relied on the fact that the cells can be synchronized by serum starvation […]" to deal with the photobleaching problem. However, this assumption may not be true as transcription factories are unsynchronous in nature and cells are also in different states and cell-cycle stages. I think the authors need to either conduct experiments to strengthen the claim and expand the discussion.

6) A general weakness of the manuscript is that the statistical significance of the data is not clearly presented. In several main figures, only single events are presented and there is no presentation of convincing statistics, such as Figure 2 and Figure 4. Where error bars are given for some data, it is unclear what are the number of events and number of independent experiments used to derive the error bars, such as Figure 5 and Figure 6. In Figure 3, where the two populations of clusters with different molecule numbers is presented, the statistics is low, in particular for the population of clusters with larger molecule numbers (only three such clusters are shown).

7) Interesting explanations for these novel observations are provided in the paper. However, the authors may want to entertain the speculation that perhaps the Pol II cluster could be a mechanistic consequence of a transient interaction with enhancers. This interaction could provide a full complement activation signals that might act on both pause release, the clustered recruitment of Pol II, and the burst of transcription.

Could this clustering of Pol II at a locus be related to the Maxwell et al. observation (Cell Rep. 2014 February 13; 6: 455-466, PMC4026043) of Pol II in an upstream "docking site" on some genes? The ChIP-seq analysis used to report docking might capture clustering events under the special regulatory state explored in that paper. I think this could be an interesting point of discussion.

---

## [Author Response]

1) The authors showed the presence of two types of Pol II clusters with distinct sizes, one with a substantially larger number of Pol II molecules than the other. They conclude that the smaller clusters represent elongating polymerase molecules on a single gene and the larger clusters present a distinct state, which is the focus of this paper. While these suggestions are reasonable, they lack experimental validation. In particular, there is no experimental evidence convincingly support that the smaller clusters represent elongating polymerase molecules on a single gene. The authors should provide such a proof which we believe could be completed within the two months we allow for return of a revised manuscript. Once this is proven, it will be clear that the clusters with a much larger molecule number represent a state distinct from elongating polymerase.

We agree that our interpretation of the function of the two types of Pol II accumulations requires a stronger experimental validation. First, our data after treatment with DRB or flavopiridol (Figure 6 and Figure 6—figure supplement 1, Figure 6—figure supplement 2) provides strong evidence for the fact that large clusters do not represents elongating Pol II molecules. DRB and flavopiridol treatments lead to a global depletion of elongating Polymerases from gene bodies (Ni et al., 2008, Rahl et al., 2010), and even after disappearance of the nascent transcripts at the gene locus, the large Pol II clusters did not disappear (Figure 6 and Figure 6—figure supplement 1) suggesting the large clusters do not represent elongating polymerases on the gene body.

A possible demonstration that small accumulations of Pol II counts at the Actb locus correspond to elongating polymerases would be to include antibody stains against Pol II phosphorylation marks specific to the elongating state (Ser2ph) in our fixed cell experiments (thus simultaneously imaging Pol II clusters, Actb mRNA, and Ser2ph). Unfortunately, such experiments are currently not feasible for us because of spectral limitations (Dendra2-Pol II occupies the blue, green and orange-red channels; Actb-MS2-JF647 occupies the far red channel).

To validate that smaller accumulation of Pol II observed in the fixed cells do indeed correspond to elongating Polymerases, we therefore set out instead to measure the correlation between Pol II cluster size and the number of nascent Actb mRNAs at individual loci in fixed cells. If small clusters corresponded to elongating polymerases, we would expect that the number of Pol II molecules (estimated with our quantitative PALM technique) would closely match the number of nascent mRNAs. This is indeed the case in the regime of low Pol II counts (up to ~20 Pol II molecules at a given allele, Pearson’s coefficient=0.63), demonstrating that small number of Pol II molecules indeed correspond to elongating polymerases. Our experiment also shows that the correlation breaks down for larger clusters (>50 Pol II molecules), confirming that large Pol II clusters are qualitatively different from the small numbers of elongating polymerases. Together, these two pieces of evidence validate our interpretation of the clusters based on their size. We have added the data as Figure 3—figure supplement 1.

*2) The authors use comparison of the apparent cluster lifetimes between live and fixed cells to conclude that the clusters are dynamic and transient in live cells. This is suggestive but not conclusive, since the photoswitch times in live and fix cells could be different. A more convincing demonstration that the transient cluster lifetime is due to transient cluster dynamics rather than photobleaching of the fluorescent proteins is to measure whether the apparent lifetime is dependent on laser intensity.*

We appreciate your suggestion of this experiment, which we have now added to the manuscript (Figure 1—figure supplement 3). We varied the excitation laser power from 20% to 150% relative to the laser power used in our initial experiments (3.2kW/cm^2^), and observed that the apparent cluster lifetime was unchanged consistent with the fact that the lifetime is not governed by photophysics but the underlying biology.

We observed that while the cluster lifetime is unchanged, the number of localizations (burst size) per clusters decreased with increasing laser power (Figure 1—figure supplement 3). The dependence of the number of localizations on the laser power reflects the fact that each fluorophore photobleaches more rapidly with higher laser power, contributing less frame counts to the cumulant trace. Representative traces of transient clusters in live cells are shown in Figure 1—figure supplement 3. Cumulant graphs and zoomed-in graphs (Figure 1—figure supplement 3) show a notable decrease of total localizations before cluster disassembly, while the lifetime of the cluster is maintained.

3) Results: "We interpret the inflection points with sudden transitions from very high to very low frequency of detections as cluster disassembly events." It is so not precise to define a plateau of a cluster's time trace as a cluster disassembly event. How is a pausing event distinguished from a disassembly event in tcPALM? For the same reason, it is not precise to define the abrupt slope as the cluster lifetime in the case that the plateau is caused by pausing of Pol II recruitment.

We realize we did not offer a clear enough explanation of our interpretation of the timetraces. We have improved the text in this respect, and are adding a supplementary figure to further illustrate this important point (Figure 1—figure supplement 2). In the new figure, we use our experimentally determined photophysical parameters to simulate the traces we would obtain in two distinct biological scenarii: short lived clusters and long-lived clusters. These simulations illustrate very clearly that 1) if the clusters were long-lived, we would observe a sustained increase in the cumulated detections over time (Figure 1—figure supplement 2), because in our regime, only a small fraction of Pol II molecules (a few percent) can be activated and photobleached in the ~10s of a cluster lifetime. 2) The onset of the plateau after each cluster gives a faithful estimate of the disassembly of the cluster (Figure 1—figure supplement 2); 3) if clusters were pre-formed prior to our experiment, we would detect them immediately. We therefore conclude that albeit relying on stochastic activations, our technique provides an accurate measurement of the start- and end-points of the clusters. The nature of the biological event driving cluster disassembly remains to be explored (for instance a polymerase escaping a paused state might very well provide the disassembly signal, see summary below and discussion of the revised manuscript), but our technique can faithfully determine when disassembly happens: if many Pol II molecules remained at the locus for longer than a few seconds, our traces would not end as abruptly as they do.

In summary, the plateau cannot be explained solely by a paused polymerase. Let us imagine a locus where a single Pol II pauses without disassembly of the cluster: since our activation power is low, we would detect a persisting slope in the tcPALM cumulant time trace, corresponding to the molecules in the cluster gradually turning on one by one (it would take minutes to entirely photoconvert a typical cluster of ~100 molecules). After minutes, the cumulant time trace would slowly saturate, as most Pol II molecules in the cluster have been photoconverted. This is indeed the case in the DRB and Flavopiridol live cell data where pausing escape is inhibited yet a slope still persists (Figure 6—figure supplement 1, and time traces in Figure 1—figure supplement 1), and illustrated in our simulations of a long-lived cluster (Figure 1—figure supplement 2).

We note that the slope in the tcPALM cumulant time trace constitutes an instant readout of the concentration of unconverted Dendra2-Pol II (this is illustrated in Figure 1—figure supplement 2 where the saturation of the bottom cumulant traces closely matches the disappearance of the green unconverted species). As a consequence, if proteins were to gradually accumulate at the locus, the cumulant time trace slope would not be constant but increase over time. This is because we typically photoconvert much slower than proteins accumulate.

In fixed cells (i.e. static clusters), the clusters are static and exist even before the start of photoactivation/recording. As such, in fixed cells data, we observe a constant slope in the cumulant time trace from the beginning of the acquisition, up to the point when most molecules have been photoconverted and the curve gradually transitions into a plateau. In fixed cells this gradual slope decrease is interpretable as the gradual depletion of the pool of photoconvertible proteins in the cluster. In live cells, by contrast, slope onset is sparse and occurs abruptly, suggesting de-novo, rapid cluster assembly, and the plateaus are also sudden, suggesting disassembly is rapid as well. The exception is in the drug-treated living cells where clusters cumulant timetraces become comparable to the fixed cells controls, suggesting that tcPALM is capable of resolving static clusters if they occur in live cells.

We therefore conclude that our drug treatment experiments suggest that blocking a Pol II in the promoter-proximal paused state (i.e. by preventing phosphorylation) also prevents cluster disassembly. This implies that cluster disassembly may be mediated by the same phosphorylation mechanism that mediates pause release.

4) Comparing Figure 5 and Figure 5—figure supplement 1, why does the rate of detection (clustering strength) remain nearly a constant while Pol II cluster lifetime has a significant increase between 10-15 mins after serum stimulation. What does longer lifetime exactly mean, number of pre-existing Pol II, fast recruitment of Pol II or slow dissociation of Pol II? And why does lifetime increase upon serum stimulation?

In tcPALM, the slope of the cumulant time trace (clustering strength) is an instant readout of the local concentration of unconverted proteins (see answer to previous comment). This is exemplified in Figure 1—figure supplement 2 in which we simulate what we would observe if we performed our experiment on a static cluster. As Pol II molecules gradually disappear (fraction of green shrinks in top left panel), the slope of the cumulant time trace slowly decreases (slope of example clusters in bottom left panels). In living cells, because clusters are short lived, we usually only convert a small fraction of the molecules within the few seconds of a clustering event (e.g. ~10 molecules out of a 100 molecules cluster within 10s Figure 1—figure supplement 2). Therefore, the lifetime we measure (the duration of the positive slope events in bottom right panel of Figure 1—figure supplement 2) reflects the duration of the biological assembly of multiple Pol II molecules. If a cluster had a longer lifetime (but the same total number of proteins), we would expect to detect molecules for a longer interval. As a result, the total amplitude of the step in the cumulant trace would be proportionally larger. However, the slope of the cumulant timetrace during the cluster would be identical. This is what we observe in our simulations: compare the cumulant time trace slope in the case of the short (10s) cluster (Figure 1—figure supplement 2, bottom right panel) to that of a longer-lived cluster (first 20s of bottom right panel in Figure 1—figure supplement 2).

*5) Results: The authors claimed that "Alternatively, we relied on the fact that the cells can be synchronized by serum starvation. […]" to deal with the photobleaching problem. However, this assumption may not be true as transcription factories are unsynchronous in nature and cells are also in different states and cell-cycle stages. I think the authors need to either conduct experiments to strengthen the claim and expand the discussion.*

We apologize for the poorly written sentence. The transcriptional activation of cells after serum starvation is stereotyped, and used to average data from multiple cells along a global activation time course. So, we relied on synchronization to average multiple cells, not to synchronize individual transcription or clustering events. We have added a note to that effect in the discussion, and revised the text to read:

“The transcriptional activation of the β-actin gene in MEF cells after serum starvation has a well-characterized stereotyped response that remains evident after averaging multiple cells (Lionnet et al., 2011a, Femino et al., 1998, Oleynikov and Singer, 2003, Kalo et al., 2015). Here we average data from individual cells selected at varying times after serum stimulation, and each cell is imaged for a short period of up to 5,000 frames (300s).”

6) A general weakness of the manuscript is that the statistical significance of the data is not clearly presented. In several main figures, only single events are presented and there is no presentation of convincing statistics, such as Figure 2 and Figure 4. Where error bars are given for some data, it is unclear what are the number of events and number of independent experiments used to derive the error bars, such as Figure 5 and Figure 6. In Figure 3, where the two populations of clusters with different molecule numbers is presented, the statistics is low, in particular for the population of clusters with larger molecule numbers (only three such clusters are shown).

We have now indicated statistical values in the manuscript and figure legends, and also provided an excel file of statistics following the *eLife* publication policy. Specifically, we additionally show examples for Figure 2 and Figure 4 (in Figure 4—figure supplement 2). We also increased our statistics, using a 4 times larger dataset for Figure 3, confirming that we consistently observe large clusters distinguished from small ones. (Now Figure 3 shows 15 large clustering events among 44 co-localizations from 34 cells.)

7) Interesting explanations for these novel observations are provided in the paper. However, the authors may want to entertain the speculation that perhaps the Pol II cluster could be a mechanistic consequence of a transient interaction with enhancers. This interaction could provide a full complement activation signals that might act on both pause release, the clustered recruitment of Pol II, and the burst of transcription.

Could this clustering of Pol II at a locus be related to the Maxwell et al. observation (Cell Rep. 2014 February 13; 6: 455-466, PMC4026043) of Pol II in an upstream "docking site" on some genes? The ChIP-seq analysis used to report docking might capture clustering events under the special regulatory state explored in that paper. I think this could be an interesting point of discussion.

This is indeed a tantalizing possibility. We have added a mention to that effect in our Discussion. That section of the Discussion now reads:

“The combination of our quantitative single cell imaging approach with genome-wide analyses can help uncover the hidden mechanisms behind clustering dependent gene expression regulation. […] Future studies bridging quantitative single cell imaging of Pol II recruitment with genome-wide mapping of Pol II occupancy and long-distance regulatory interactions will provide important insights in the relationships between chromatin architecture, docking, pausing/release, and Pol II clustering.”